# Topographic design in wearable MXene sensors with in-sensor machine learning for full-body avatar reconstruction

Haitao Yang[1,9], Jiali Li[1,9], Xiao Xiao [2,9], Jiahao Wang[3,9], Yufei Li[1], Kerui Li[1], Zhipeng Li [3], Haochen Yang [4], Qian Wang[1], Jie Yang [1], John S. Ho [3], Po-Len Yeh[5], Koen Mouthaan[3], Xiaonan Wang [6], Sahil Shah [7] ✉ & Po-Yen Chen [4,8] ✉

Wearable strain sensors that detect joint/muscle strain changes become prevalent at human–machine interfaces for full-body motion monitoring. However, most wearable devices cannot offer customizable opportunities to match the sensor characteristics with specific deformation ranges of joints/muscles, resulting in suboptimal performance. Adequate wearable strain sensor design is highly required to achieve user-designated working windows without sacrificing high sensitivity, accompanied with real-time data processing. Herein, wearable $Ti_3C_2T_x$ MXene sensor modules are fabricated with in-sensor machine learning (ML) models, either functioning via wireless streaming or edge computing, for full-body motion classifications and avatar reconstruction. Through topographic design on piezoresistive nanolayers, the wearable strain sensor modules exhibited ultrahigh sensitivities within the working windows that meet all joint deformation ranges. By integrating the wearable sensors with a ML chip, an edge sensor module is fabricated, enabling in-sensor reconstruction of high-precision avatar animations that mimic continuous full-body motions with an average avatar determination error of 3.5 cm, without additional computing devices.

Tracking, monitoring, and reconstruction of full-body motions are increasingly prevalent and have been applied in many scenarios, including high-precision movement detection[1,2], motor sign recognition[3], athlete performance analysis[4,5], rehabilitation assessment[6,7], human–machine interaction[8,9], and personalized avatar reconstruction in augmented/virtual reality[10,11]. Current approaches to realizing full-body motion monitoring involve digital imaging systems (such as cameras), which capture a series of photographs and/or videos to extract quantitative movement information[12,13]. However, the approaches of using digital imaging systems for body motion monitoring face several limitations. A key drawback is that these imaging systems are composed of immobile, expensive, and burdensome equipment, making them hard to relocate and unsuitable for tracking distant and dynamic objects[1,14]. Also, privacy and data security concerns may constrain the implementation of cameras in communities or home settings[15,16]. Furthermore, graphics processing units (GPUs) are

[1]Department of Chemical and Biomolecular Engineering, National University of Singapore, 4 Engineering Drive 4, Singapore 117585, Singapore. [2]Department of Electrical and Electronic Engineering, Southern University of Science and Technology, Shenzhen, China. [3]Department of Electrical and Computer Engineering, National University of Singapore, Singapore 117583, Singapore. [4]Department of Chemical and Biomolecular Engineering, University of Maryland, College Park, MD 20740, USA. [5]Realtek, Singapore 609930, Singapore. [6]Department of Chemical Engineering, Tsinghua University, 100084 Beijing, China. [7]Department of Electrical and Computer Engineering, University of Maryland, College Park, MD 20740, USA. [8]Maryland Robotics Center, College Park, MD 20740, USA. [9]These authors contributed equally: Haitao Yang, Jiali Li, Xiao Xiao, Jiahao Wang. ✉e-mail: sshah389@umd.edu; checp@umd.edu

always needed to process images/videos in external data terminals[17,18], proposing the challenges of high bandwidth requirement, hardware expense, and power consumption.

An alternative strategy for high-precision full-body motion monitoring and/or avatar reconstruction involves wearable strain sensors that can mechanically conform to dynamic joint surfaces of the human body, allowing the physiological signals to be collected[1,4,6,8,19,20]. For proprioception purposes, the multi-joint motions of the human body require a set of wearable strain sensors capable of achieving high sensitivities in separate strain ranges[8,21,22]. However, the state-of-the-art strain sensors and existing commercial sensors have limited customizable opportunities to tune the sensors' working windows to match the strain changes of targeted joints/muscles[7,23–25], leading to erroneous sensing signals and low signal-to-noise ratios. Therefore, to satisfy full-body motion monitoring applications, adequate wearable strain sensor design is highly required to achieve user-designated working windows without sacrificing high sensitivity, accompanied with real-time data processing.

Besides tuning the strain sensor characteristics, an additional challenge is to transmit, store, and process the raw sensor data collected through multiple signal acquisition channels[7,21,26]. One facile approach is through the integration of wireless technology (e.g., Bluetooth) with wearable strain sensors, enabling continuous, real-time streaming of multi-channeled sensor data to an external computing device[26–28]. On the other hand, an emerging approach is to process the time-resolved sensor data locally (in-sensor), which can largely reduce communication bandwidths and radio power consumption as well as improve data latency and security[28,29]. To the best of our knowledge, integrating edge computing chip(s) with wearable strain sensors (with customizable sensor characteristics) has not been realized in the literature, especially to monitor and analyze the multi-joint and multi-mode movements of the human body.

Herein, wearable sensor modules were fabricated with ML models, either functioning via wireless data streaming or in-sensor edge computing, for full-body motion classifications and personalized avatar reconstruction. $Ti_3C_2T_x$ MXene nanosheets were specifically adopted for the fabrication of piezoresistive nanolayers due to high electrical conductivity, superior conformability, and ease of processing[30–34]. First, by harnessing the interfacial instability during localized thermal contraction, wrinkle-like topographies were heterogeneously created on the piezoresistive MXene nanolayers, the crack propagation behaviors of which were able to be controlled. Through topographic design, the working windows of resulting strain sensors were managed to be tuned from 6 to 84%, which met the strain ranges of all the joints without sacrificing ultrahigh sensitivities (gauge factor (GF) > 1000). Next, by interfacing wearable strain sensors with Bluetooth chips, the wireless sensor module was able to stream multi-channeled strain sensing data continuously, which were input to train an Artificial Neural Networks (ANN) model capable of identifying a variety of full-body motions with 100% classification accuracy. Finally, an edge sensor module was fabricated by integrating wearable strain sensors with an edge computing chip, enabling in-sensor Convolutional Neural Network (CNN) to reconstruct personalized avatar animations with an average avatar determination error of 3.5 cm. In comparison with the wireless sensor module, the edge sensor module avoided wireless data streaming and showed 71% less power consumption for full-body avatar reconstruction.

## Results

### Controllable fabrication of wrinkle-like MXene textures via localized thermal contraction

A variety of low-dimensional nanomaterials, ranging from one-dimensional (1D) silver nanowires[35,36] and carbon nanotubes[37,38] to two-dimensional (2D) graphene[39,40] and $Ti_3C_2T_x$ MXene nanosheets[7,41,42], have been adopted for the fabrication of piezoresistive nanolayers.

Polymers are often involved to stabilize the assembled nanostructures[30,43]. In this work, three building block units, single-walled carbon nanotubes (SWNTs), $Ti_3C_2T_x$ MXene nanosheets, and polyvinyl alcohol (PVA), were selected for the fabrication of piezoresistive nanolayers. $Ti_3C_2T_x$ MXene nanosheets were specifically selected over other 2D materials because of their high electrical conductivities, superior mechanical properties, intrinsic hydrophilicity, and ease of processing[30–34]. Detailed characterizations of SWNTs and MXene nanosheets are provided in Supplementary Fig. 1, where the diameter and length of SWNTs were 1–2 nm and 5–30 μm, respectively, and the average diameter of as-exfoliated MXene nanosheets was characterized to be ca. 1000 nm.

Next, the dispersions/solution of MXene nanosheets, SWNTs, and PVA were mixed and then underwent vacuum-assisted filtration to deposit a composite nanolayer on a polyvinylidene fluoride (PVDF) membrane. The thickness of as-filtered MXene/SWNT/PVA nanolayer (abbreviated as ps-MXene nanolayer) was fixed at 400 nm for the rest of this study, and the composition of ps-MXene nanolayer was controllable by adjusting the mass ratios of three building block units in the mixture. Supplementary Fig. 2a–c present the top-down and cross-section scanning electron microscope (SEM) images of a planar ps-MXene nanolayer, where SWNTs were highly dispersed and entangled within the MXene multilayer. Further characterizations of ps-MXene nanolayers (including X-ray diffraction (XRD) and Raman spectra) are provided in Supplementary Fig. 2d, e, respectively. The as-filtered ps-MXene nanolayer was then detached from the PVDF membrane in an ethanol bath, and the freestanding ps-MXene nanolayer was transferred onto a thermally responsive polystyrene (PS) shrink film. In this study, the shrink films were specifically customized to contract only in a uniaxial direction above the glass transition temperature ($T_g$) of PS (ca. 100 °C), which were produced by using the roll-to-roll machine shown in Supplementary Fig. 3. After the thermal contraction at 100 °C for 120 s, the width of the uniaxial shrink film was reduced 50%, while its length remained constant (see Supplementary Figs. 4 and 5 for detailed characterizations).

By controlling the thermal contraction region(s), the planar ps-MXene nanolayers (named as $M_p$, $M$ indicates the ps-MXene nanolayer, the subscript $p$ refers to the planar feature) were managed to be deformed into different kinds of $M_n$ nanolayers (the subscript $n$ refers to the topographic design). Two categories of $M_n$ nanolayers were developed in this work: (1) the ps-MXene nanolayers with homogenous topographies (including $M_p$ and $M_w$) and (2) the ps-MXene nanolayers with heterogeneous topographies (including $M_{p-w-p}$ and $M_{w-p-w}$).

In the category of homogeneous topographies, a $M_p$ nanolayer was shrunk without any constraints, and the resulting nanolayer was fully covered with periodic wrinkles, which was named as $M_w$ nanolayer (the subscript $w$ refers to the wrinkle-like feature, Fig. 1a, (1)). Supplementary Figs. 6 and 7 show the digital photos and SEM images of a $M_w$ nanolayer, respectively. In the category of heterogenous topographies, the two ends of a $M_p$ nanolayer were fixed (Fig. 1a, (2)), so only the middle region showed periodic wrinkles after thermal contraction, which was named as $M_{p-w-p}$ (the subscript $w$ refers to the wrinkle-like feature, $p$ refers to the planar feature, Fig. 1a, (2)). When the middle part of a $M_p$ nanolayer was fixed, the two edge regions were shrunk and displayed periodic wrinkles (named as $M_{w-p-w}$ nanolayer, the subscript $w$ refers to the wrinkle-like feature, $p$ refers to the planar feature, Fig. 1a, (3)). Figure 1b presents the SEM images of $M_w$, $M_{p-w-p}$, and $M_{w-p-w}$ nanolayers, showing that the generation of ps-MXene micro-wrinkles can be locally controlled. Supplementary Fig. 8 further shows the enlarged SEM image of the transition regions of $M_{p-w-p}$ and $M_{w-p-w}$ nanolayers (between planar and textured regions). The wavelength distribution profiles of $M_p$, $M_w$, $M_{p-w-p}$, and $M_{p-w-p}$ nanolayers are summarized in Fig. 1c, showing that the periodic wrinkles are constrained within the region(s) experiencing localized thermal contraction. The electrical conductivity of a $M_p$ nanolayer (at the MXene/

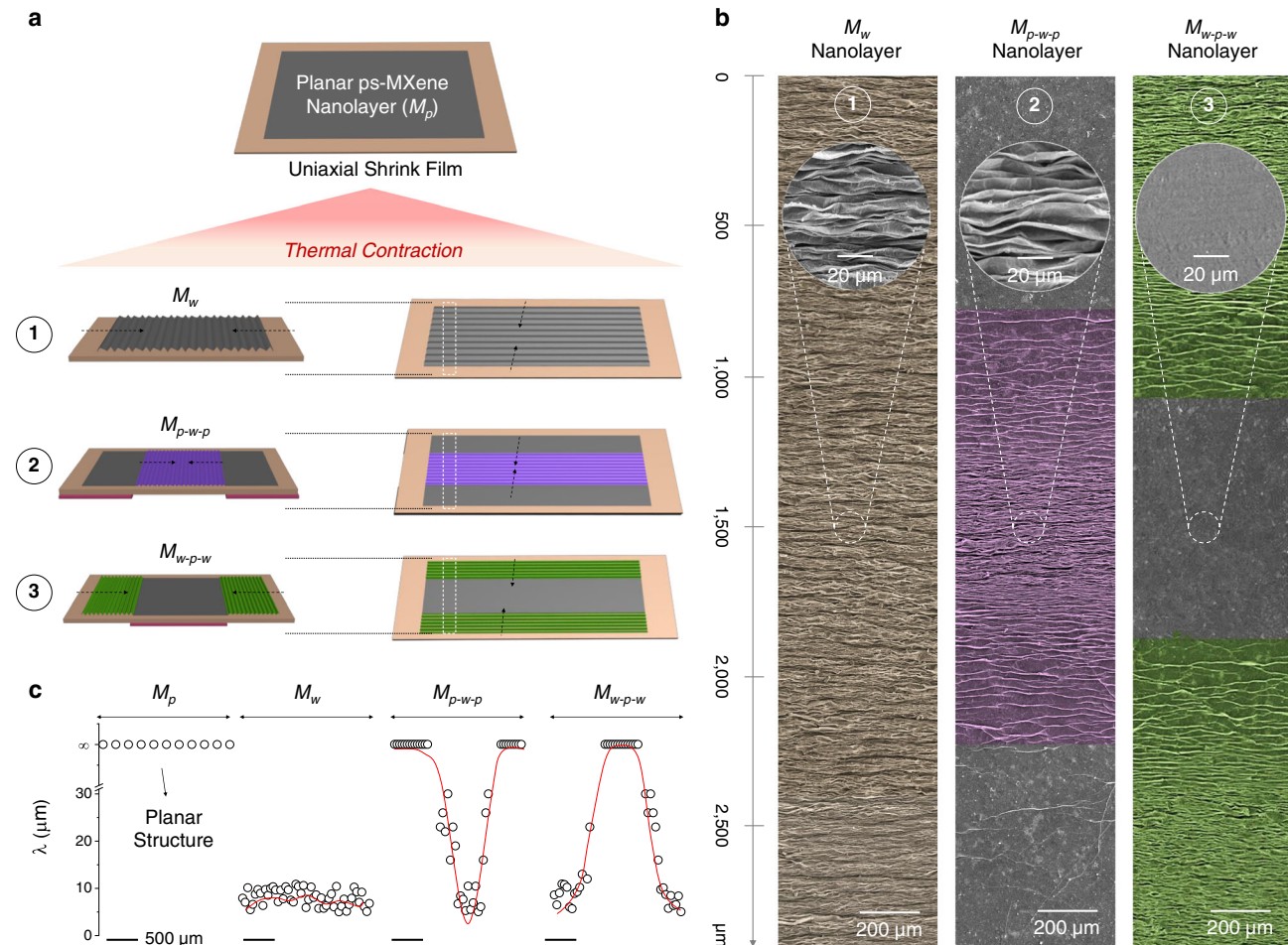

**Fig. 1 | Controllable fabrication of wrinkle-like MXene textures via localized thermal contraction. a** Fabrication of $M_w$, $M_{p\text{-}w\text{-}p}$, and $M_{w\text{-}p\text{-}w}$ nanolayers with localized periodic wrinkles. The red color regions refer to the areas fixed with thin glasses. **b** SEM images of $M_w$, $M_{p\text{-}w\text{-}p}$, and $M_{w\text{-}p\text{-}w}$ nanolayers. **c** Wavelength distribution profiles of $M_p$, $M_w$, $M_{p\text{-}w\text{-}p}$, and $M_{w\text{-}p\text{-}w}$ nanolayers. Scale bars: 500 μm.

SWNT/PVA ratio of 85/10/5) was confirmed to be 2479 S cm$^{-1}$ using a four-point probe, and the electrical resistances of all kinds of wrinkle-like ps-MXene textures were provided in Supplementary Fig. 9.

## Crack propagation behaviors of $M_n$ sensors with homogenous and heterogenous topographies

For the fabrication of wearable strain sensors, all the $M_n$ nanolayers (including $M_p$, $M_w$, $M_{p\text{-}w\text{-}p}$, and $M_{w\text{-}p\text{-}w}$) were first immersed in a dichloromethane (DCM) bath to dissolve the PS substrates, and the $M_n$ nanolayers were detached and became freestanding (see Supplementary Fig. 10 and *Experimental Section* for fabrication details). As shown in the SEM image in Supplementary Fig. 11 and the atomic force microscopy (AFM) profiles in Supplementary Fig. 12, the wrinkle-like textures of a $M_w$ nanolayer were slightly relaxed, and the average wavelength increased from 8.1 to 10.2 μm. Next, the freestanding $M_n$ nanolayers were transferred onto VHB™ tapes followed by wiring electrical leads to obtain the corresponding $M_n$ sensors (as illustrated in Supplementary Figs. 13 and 14).

Finite element analysis (FEA) was conducted to simulate the strain distribution heatmap of each $M_n$ sensor under uniaxial strains. With homogenous topographies, the FEA results of $M_p$ and $M_w$ sensors were compared. As shown in Fig. 2a, i, when the $M_p$ sensor was stretched to 120% of its original length, the $M_p$ nanolayer experienced large and concentrated strains (>50%) in its central region. In comparison, in Fig. 2a, ii, when the $M_w$ sensor was stretched to 120%, the $M_w$ nanolayer experienced attenuated localized strains (<20%), and the localized

strains were propagated at the valleys of periodic wrinkles. The crack propagation behaviors of $M_p$ and $M_w$ sensors during continuous strain loading processes were in situ recorded by using a reflection-contrast microscopy. As recorded in Fig. 2b and Supplementary Movie 1, when the $M_p$ sensor was stretched to 120%, long and continuous fractures were generated on its piezoresistive nanolayer. On the other hand, when the $M_w$ sensor was stretched to 120%, short and zigzag surface cracks were gradually developed (see Fig. 2b and Supplementary Movie 2). As further shown in the SEM images in Supplementary Figs. 15 and 16, the $M_w$ nanolayer showed slower crack propagation than the $M_p$ nanolayer under stretching, preventing the conductive pathways of nanolayer from being completely cut off.

With heterogenous topographies, the FEA results of $M_{p\text{-}w\text{-}p}$ and $M_{w\text{-}p\text{-}w}$ sensors were compared. As shown in Fig. 2c, iii, the $M_{p\text{-}w\text{-}p}$ sensor showed the strain distribution profile with hybrid features, where the wrinkle-textured regions experienced attenuated strains (<20%) while the localized strain in the planar region quickly propagated above 50%. Similarly, in Fig. 2c, iv, the $M_{w\text{-}p\text{-}w}$ sensor presented a region-dependent strain distribution profile. As a result, in Fig. 2d, both $M_{p\text{-}w\text{-}p}$ and $M_{w\text{-}p\text{-}w}$ sensors showed region-dependent crack propagation behaviors during the strain loading processes: long/continuous fractures emerged in the planar regions, and short/zigzag cracks propagated in the textured regions. By using *ImageJ* to quantify the FEA results under 120% stretching, the $M_{w\text{-}p\text{-}w}$ nanolayer showed an average localized strain of 20%, lower than the $M_{p\text{-}w\text{-}p}$ nanolayer (32%). Therefore, the $M_{w\text{-}p\text{-}w}$ nanolayer showed slower crack propagation than the $M_{p\text{-}w\text{-}p}$ nanolayer

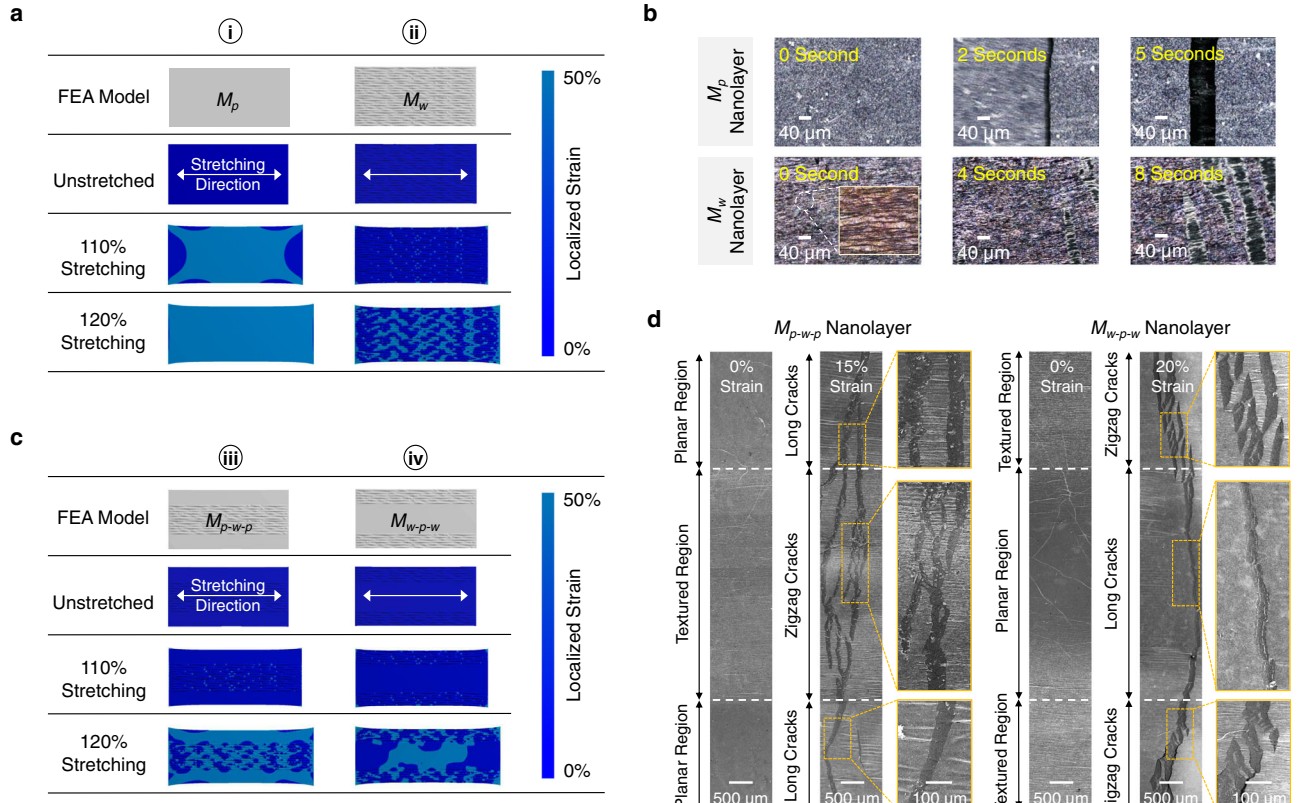

**Fig. 2 | Crack propagation behaviors of $M_n$ sensors with homogenous and heterogenous topographies. a** FEA simulations of strain distribution profiles of $M_p$ and $M_w$ nanolayers. **b** Reflection-contrast microscopy images of $M_p$ and $M_w$ sensors for in situ observation of their crack propagation behaviors. **c** FEA simulations of strain distribution profiles of $M_{p-w-p}$ and $M_{w-p-w}$ nanolayers. **d** SEM images of $M_{p-w-p}$ and $M_{w-p-w}$ sensors under uniaxial strains for in situ observation of their crack propagation behaviors.

(see SEM images in Supplementary Figs. 17 and 18), enabling $M_{w-p-w}$ sensors to sustain higher uniaxial strain (>50%) than the $M_{p-w-p}$ sensor (<30%) before the conductive pathways of nanolayer being completely cut off. From both FEA simulation and experimental results, the crack propagation behaviors of $M_n$ sensors can be controlled by creating homogenous and heterogeneous topographies.

## Tunable strain sensing characteristics of $M_n$ sensors through topographic designs and stretching directions

Three characteristics of a strain sensor are generally evaluated, including (1) sensitivity, (2) linear working window, and (3) maximum working strain. The sensitivity of a strain sensor is normally characterized by GF, as defined in Eqs. 1 and 2,

$$S_\varepsilon = \frac{R_\varepsilon - R_0}{R_0} \tag{1}$$

$$GF = \frac{S_\varepsilon}{\varepsilon} \tag{2}$$

where $S_\varepsilon$ is the relative resistance change at $\varepsilon$ strain, $\varepsilon$ denotes the applied strain, $R_0$ and $R_\varepsilon$ represent the initial resistance and the resistance under $\varepsilon$ strain, respectively. On the other hand, the linear working window of a strain sensor is determined by the strain range where its resistance increased linearly with the applied strain. When the strain sensor reaches its maximum resistance, the maximum working strain is abbreviated as $\varepsilon_{max}$.

As the $M_n$ nanolayers exhibited directional wrinkle-like textures, the stretching direction showed significant effects on the strain sensing characteristics of a $M_n$ sensor. Figure 3a, b present the relative

resistance change ($S_\varepsilon$)−strain ($\varepsilon$) profiles of $M_n$ sensors under two stretching directions (parallel to and perpendicular to the wrinkle axes). Under parallel stretching (Fig. 3a), the linear working windows of $M_p$, $M_w$, $M_{p-w-p}$, and $M_{w-p-w}$ sensors were characterized to be 3–6%, 8–24%, 25–39%, and 35–50%, respectively, with high GF values of 3400, 1160, 1230, and 1470. Also, the $\varepsilon_{max}$ of $M_p$, $M_w$, $M_{p-w-p}$, and $M_{w-p-w}$ sensors were determined to be 6%, 24%, 39%, and 50%, respectively. In comparison, under perpendicular stretching (Fig. 3b), the $M_w$, $M_{w-p-w}$, $M_{p-w-p}$ sensors demonstrated narrower linear working windows (with an average strain range of 6%) and lower sensitivities (GF ~600). In Supplementary Note 1, detailed FEA simulations and in situ SEM studies were conducted on all $M_n$ sensors to investigate the effect of stretching directions on their crack propagation behaviors and strain sensing performance. In short, there are two major advantages of selecting parallel stretching over perpendicular stretching, including (1) higher $M_n$ sensors' sensitivities and wider linear working windows and (2) more design opportunities via topographic design. Therefore, we adopted the parallel stretching direction for all $M_n$ sensors. Detailed comparisons of strain sensor performance between (1) $M_p$ and $M_w$ sensors and (2) $M_{p-w-p}$ and $M_{w-p-w}$ sensors are supplemented in Supplementary Note 2.

To further examine the influence of the topographic design on the strain sensing performance, several types of $M_n$ sensors were fabricated by (1) varying the areal percentages of wrinkle-like region(s) and (2) changing the distribution of wrinkle-like region(s). First, Supplementary Fig. 19 compares the strain sensing curves of various $M_{w-p-w}$ sensors with different areal percentages of wrinkle-like regions. With the areal percentages of wrinkle-like regions increasing from 5 to 75%, the resulting $M_{w-p-w}$ nanolayers exhibited more short/zigzag cracks under parallel stretching. The higher

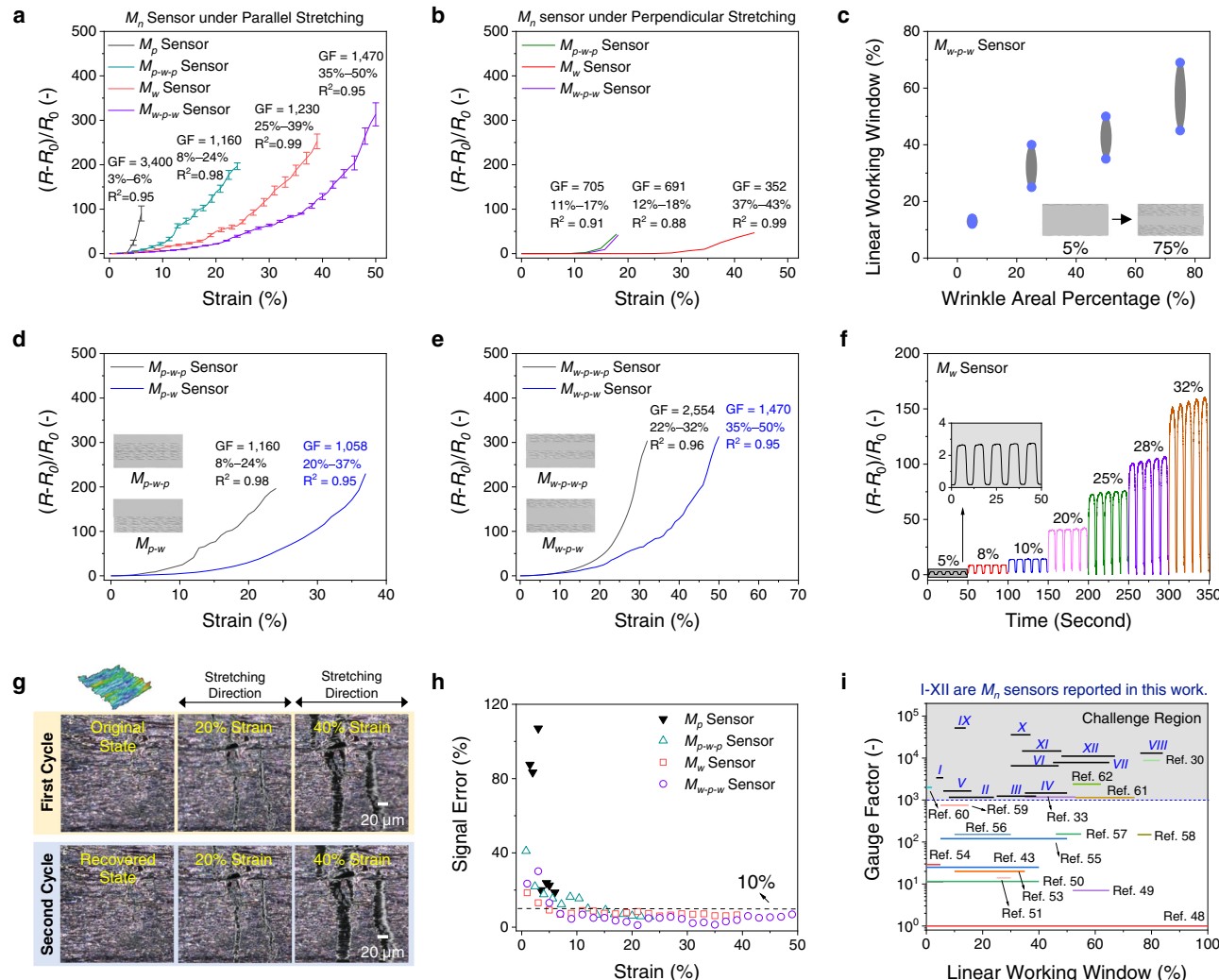

**Fig. 3 | Tunable strain sensing characteristics of $M_n$ sensors through topographic designs and stretching directions. a** Relative resistance change ($S_\varepsilon$)–strain ($\varepsilon$) curves of $M_n$ sensors under parallel stretching. Error bar refers to the standard deviations, which were calculated based on three replicates. The composition of all $M_n$ nanolayers was set at 85/10/5 (MXene/SWNT/PVA), and the thickness of all $M_n$ nanolayers was controlled at 400 nm. **b** Relative resistance change ($S_\varepsilon$)–strain ($\varepsilon$) curves of $M_n$ sensors under perpendicular stretching. **c** Linear working window(s) of the $M_{w\text{-}p\text{-}w}$ sensors with different areal percentages of wrinkle-like region(s). **d** Strain sensing curves of $M_{p\text{-}w\text{-}p}$ and $M_{p\text{-}w}$ sensors. **e** Strain sensing curves of $M_{w\text{-}p\text{-}w}$ and $M_{w\text{-}p\text{-}w}$ sensors. **f** Time-dependent relative resistance changes of a $M_w$ sensor under various repeated uniaxial strains. **g** Reflection-contrast microscopy images for in situ observation of structural evolutions of a $M_w$ nanolayer under repeated uniaxial strains. **h** Signal errors of $M_n$ sensors. **i** Comparison of our $M_n$ sensors with other strain sensors in the literature (in terms of gauge factor and linear working windows ($R^2 \geq 0.95$)). The gray color region represents the challenge region for current piezoresistive strain sensors to realize the gauge factors >1000. The fabrication details of 12 $M_n$ sensors are listed in Supplementary Table 1.

coverage of short/zigzag cracks prevented the conductive pathways from being completely cut off, leading to wider working windows. Therefore, as the areal percentages increased from 5 to 75%, the linear working window of the resulting $M_{w\text{-}p\text{-}w}$ sensor increased from 12–14% to 45–69%, respectively (Fig. 3c).

Second, several other types of $M_n$ sensors were fabricated by varying the positions and distribution of wrinkle-like region(s); the total areal percentage of wrinkle-like region(s) was controlled to be the same at 50%. To evaluate the effect of wrinkle texturing distribution on the strain sensing performance, two kinds of $M_n$ sensors, including $M_{p\text{-}w}$ and $M_{w\text{-}p\text{-}w}$, were fabricated. Supplementary Fig. 20 shows the FEA results of $M_{p\text{-}w}$, $M_{p\text{-}w\text{-}p}$, $M_{w\text{-}p\text{-}w}$, and $M_{w\text{-}p\text{-}w}$, and the average localized strain of each sensor was quantified by *ImageJ* on the FEA results. Figure 3d compares the strain sensing curves of $M_{p\text{-}w}$ and $M_{p\text{-}w\text{-}p}$, which have "one" wrinkle-like region. The $M_{p\text{-}w\text{-}p}$ nanolayer under 120% stretching showed an average localized strain of 32%, while the $M_{p\text{-}w}$ nanolayer illustrated a lower average localized strain of 17%. With lower localized strains, the $\varepsilon_{max}$ of a $M_{p\text{-}w}$ sensor was characterized to be 37%,

which was larger than the one of a $M_{p\text{-}w\text{-}p}$ sensor ($\varepsilon_{max}$ = 24%). Figure 3e compares the strain sensing curves of $M_{w\text{-}p\text{-}w}$ and $M_{w\text{-}p\text{-}w\text{-}p}$, which have "two" wrinkle-like regions. The $M_{w\text{-}p\text{-}w}$ nanolayer under 120% stretching showed an average localized strain of 20% (quantified by *ImageJ* on the FEA result), while the $M_{w\text{-}p\text{-}w\text{-}p}$ nanolayer illustrated a higher average localized strain of 34%. With lower localized strains, the $\varepsilon_{max}$ of a $M_{w\text{-}p\text{-}w}$ sensor was characterized to be 50%, which was higher than the one of a $M_{w\text{-}p\text{-}w\text{-}p}$ sensor ($\varepsilon_{max}$ = 32%). From both FEA results and strain sensing performance, setting the wrinkle-like region(s) at the edge position(s) was able to effectively reduce overall localized strains and increase $M_n$ sensors' $\varepsilon_{max}$.

The strain sensing stability of all $M_n$ sensors was investigated. Figure 3f demonstrates stable relative resistance changes of a $M_w$ sensor under repeated uniaxial strains, as the surface cracks were repeatedly generated at the same spots on the reconfigurable microtextures during reversible stretching processes (in situ recorded in Fig. 3g and Supplementary Movie 3). It is worth to note that the flattened plateaus on the signal peaks in Fig. 3f resulted from the default

mode of our tensile tester, as the movement of tensile grips slowed down to transit from stretching (open grips) to relaxation (close grips). From Supplementary Figs. 21–24, the cycling performance of all $M_n$ sensors was tested for 20,000 cycles, where the relative resistance changes of all $M_n$ sensors remained stable during the cycling tests. In addition, the response times of a $M_p$ sensor were collected in Supplementary Fig. 25.

The mechanical properties of all $M_n$ sensors were tested. As shown in Supplementary Fig. 26, all $M_p$, $M_{p\text{-}w\text{-}p}$, $M_w$, and $M_{w\text{-}p\text{-}w}$ sensors showed similar Young's moduli of ca. 150 kPa, which were higher than a bare VHB$^{TM}$ tape (106 kPa). In addition, the hysteresis of a $M_n$ sensor ($U_{hysteresis}$) was quantified by measuring the maximal signal difference between the stretching and releasing processes, as defined in Eq. 3,

$$U_{hysteresis} = Max|S_{stretching} - S_{releasing}| \tag{3}$$

where $S_{stretching}$ is the relative resistance change signal, $(R\text{-}R_O)/R_O$, of a $M_n$ sensor during the stretching process, and $S_{releasing}$ is the relative resistance change of a $M_n$ sensor during the relaxation process. Based on Supplementary Fig. 27, the hysteresis values of $M_p$, $M_{p\text{-}w\text{-}p}$, $M_w$, and $M_{w\text{-}p\text{-}w}$ sensors were calculated as 18, 27, 28, and 31, respectively, as the hysteresis of a VHB tape increased with the applied strains of 5%, 15%, 25%, and 40%[44,45].

Fabrication reproducibility is highly critical for wearable sensor applications, as the users can avoid re-calibration of as-fabricated $M_n$ sensors and use the in-database $S_\varepsilon$–$\varepsilon$ profiles as the references[23,46,47]. In this study, the fabrication reproducibility was characterized by calculating the signal error of three $M_n$ sensor replicates, as defined in Eq. 4,

$$Signal\ Error = \frac{\sigma_{S_\varepsilon}}{\varepsilon} \tag{4}$$

where $\sigma_{S_\varepsilon}$ is the standard deviation of $S_\varepsilon$ under an applied strain ($\varepsilon$). A smaller signal error indicates that $M_n$ sensor replicates are with nearly identical strain sensing characteristics, thus showing higher fabrication reproducibility, vice versa. The large signal error resulted from large $S_\varepsilon$ variations from three $M_n$ sensor replicates. According to Supplementary Movie 1, the $M_p$ nanolayer exhibited uncontrolled crack propagation behaviors, showing that the large cracks were randomly generated under strains. In comparison, according to Supplementary Movie 2 and 3, the $M_w$ nanolayer demonstrated a more controllable crack propagation fashion, showing that the zigzag cracks were constrained along the valleys of periodic wrinkles and emerged repeatedly under strains. Therefore, with one or more wrinkle-textured region(s), the $M_w$, $M_{w\text{-}p\text{-}w}$, and $M_{p\text{-}w\text{-}p}$ sensors (three replicates) demonstrate more consistent strain sensing performance and thus lower signal errors <10% (as shown in Fig. 3h). On the other hand, with only planar nanolayers, the $M_p$ sensors (three replicates) exhibited the largest signal errors >50%.

The effects of nanolayer composition and thickness on the performance of $M_n$ sensors were also investigated. By fixing the nanolayer composition (at the MXene/SWNT/PVA ratio of 85/10/5) and thickness (at 400 nm) (Supplementary Fig. 28a), the $\varepsilon_{max}$ of resulting $M_n$ sensors was tuned from 6 to 50%, when the nanolayer topography altered from $M_p$ to $M_{w\text{-}p\text{-}w}$. In comparison, by fixing the nanolayer thickness (at 400 nm) (Supplementary Fig. 28b), the $\varepsilon_{max}$ only increased from 6 to 16% for the $M_p$ sensors, from 24 to 47% for the $M_{p\text{-}w\text{-}p}$ sensors, from 39 to 65% for the $M_w$ sensors, and 50% to 84% for the $M_{w\text{-}p\text{-}w}$ sensors, when the nanolayer composition varied from 85/10/5 to 65/30/5. On the other hand, by fixing the nanolayer composition (at the MXene/SWNT/PVA ratio of 65/10/5) (Supplementary Fig. 28c), the $\varepsilon_{max}$ decreased from 16 to 14% for the $M_p$ sensors, from 47 to 37% for the $M_{p\text{-}w\text{-}p}$ sensors, from 65 to 48% for the $M_w$ sensors, and 84 to 67% for the $M_{w\text{-}p\text{-}w}$ sensors, when the nanolayer thickness increased from 400 to 800 nm. Supplementary Note 3 provided more discussions of the effects of

nanolayer thicknesses on the $M_n$ sensors' topographic design and crack propagation behaviors. Supplementary Fig. 29 summarizes the nanolayer composition, thickness, and topography effects on the $\varepsilon_{max}$ of $M_n$ sensors. Compared with the strategies of adjusting nanolayer composition and thickness, topographic design served as a more effective approach to tuning the $M_n$ sensor characteristics. As further summarized in Fig. 3i, the linear working windows of $M_n$ sensors (indices I–XII, see fabrication details in Supplementary Table 1) were able to be tuned from 6 to 84%. In comparison with the state-of-the-art strain sensors (summarized in Fig. 3i), our $M_n$ sensors showed user-designated linear working windows without sacrificing their superior strain sensitivities (GF > 1000)[30,33,43,48–62].

## Wireless sensor module for full-body motion classification

With tunable linear working windows, the $M_n$ sensors are suitable to match the strain changes of different body joints, and the proprioception information can be collected with high accuracy to classify, track, and reconstruct full-body motions. As shown in Fig. 4a, by measuring the average strain change ranges of seven joints of a volunteer (including back waist, left/right shoulders, left/right elbows, and left/right knees), we designed and fabricated seven $M_n$ sensors with joint-matched linear working windows. In detail, the $M_p$ sensor with the linear working window of 3–6% was selected to monitor the back waist bending (average strain change ~5%). The $M_{p\text{-}w\text{-}p}$, $M_w$, and $M_{w\text{-}p\text{-}w}$ sensors with the windows of 8–24%, 25–39%, and 35–50% were assigned to monitor the movements of shoulders (average strain change ~10%), elbows (average strain change ~30%), and knees (average strain change ~50%), respectively.

The $M_n$ sensors in a single type (e.g., all $M_p$ sensors) were not able to collect accurate proprioception signals for full-body motion monitoring. When seven $M_{w\text{-}p\text{-}w}$ sensors were attached to all the joints (Fig. 4b), the $S_\varepsilon$ signals collected from back waist were too small (<1.0) and hard to be distinguished from noises. On the other hand, when seven $M_p$ sensors were attached to all the joints (Fig. 4c–e), the average strain changes of shoulders, elbows, and knees exceeded the linear working window of $M_p$ sensor, leading to erroneous $S_\varepsilon$ signals. Similarly, using single type of $M_{p\text{-}w}$ or $M_w$ sensors was not able to monitor the high-strain movements of left/right knees (Supplementary Fig. 30). Therefore, it was necessary to use multi-type $M_n$ sensors to collect the proprioception signals of full-body motions. As shown in Fig. 4f, the seven $M_n$ sensors were able to successfully record the strain sensing signals in a multi-channeled fashion for various full-body motions, including (i) left/right elbow lifting, (ii) left/right shoulder lifting, (iii) squatting, (iv) stooping, (v) walking, and (vi) running.

To facilely transmit the multi-channeled sensor data to a terminal computing device, the seven $M_n$ sensors were interfaced with a Bluetooth chip for the fabrication of a wireless sensor module. The wireless sensor module was composed of seven $M_n$ sensors with joint-matched linear working windows, an analog-to-digital converter (ADC) with multi-data-acquisition channels, a microcontroller unit (MCU), and a Bluetooth chip. The equivalent circuit is illustrated in Fig. 4g, and the digital photo is shown in Supplementary Fig. 31. Each $M_n$ sensor was connected in series with standard resistors, and the resistor value was specifically selected to be 100 kΩ to ensure wireless data transmission with high accuracy (see Supplementary Note 4, Supplementary Fig. 32, and *Methods* for more details). By applying a constant voltage of 5.0 V ($V_{in}$), the voltage outputs ($V^i$) were measured by the ADC unit and derived in Eq. 5,

$$V^i = V_{in}\frac{R^i \times R_{ADC}}{R^i \times R_{ADC} + R_s \times (R^i + R_{ADC})} \tag{5}$$

where $V^i$ is the voltage output of $i^{th}$ sensor channel ($i = 1$–7), $V_{in}$ is the applied voltage (i.e., 5.0 V), $R_s$ is the resistance of the standard resistor

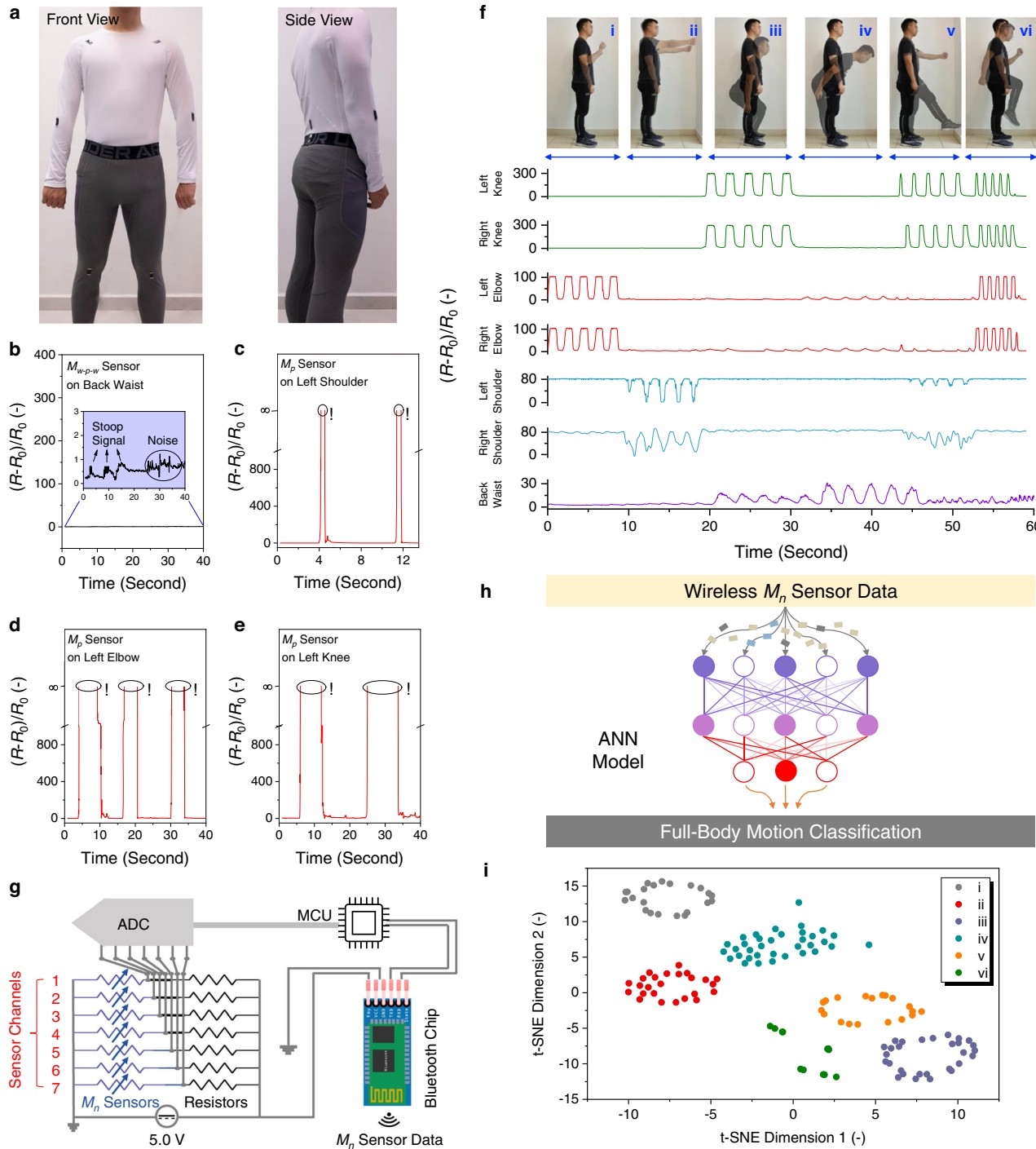

**Fig. 4 | Wireless sensor module for full-body motion classification. a** Photos of seven $M_n$ sensors attached on the back waist (one $M_p$), left/right shoulders (two $M_{p\cdot w\cdot p}$), left/right elbows (two $M_w$), and left/right knees (two $M_{w\cdot p\cdot w}$) of a volunteer. **b** Signal outputs, $S_\varepsilon$, of a $M_{w\cdot p\cdot w}$ sensor attached on the back waist during repeated stoop motions were too small to be distinguished from noise signals. Signal outputs, $S_\varepsilon$, of two $M_p$ sensors attached on (**c**) the left shoulder, (**d**) the left elbow, and (**e**) the left knee during repeated movements. Symbol "!" indicates that the $M_p$ sensors' resistances increased to infinite, where $M_p$ sensors lost their strain sensing capabilities. **f** Signal outputs, $S_\varepsilon$, of seven $M_n$ sensors for full-body motion monitoring, including (i) left/right elbow lifting, (ii) left/right shoulder lifting, (iii) squatting, (iv) stooping, (v) walking, and (vi) running. **g** Equivalent circuit of a wireless sensor module. **h** Multi-channeled $M_n$ sensor data were collected to construct a high-accuracy ANN model for full-body motion classification. **i** t-SNE scatterplot of six full-body motions, where the strain sensing data underwent the dimension reduction into two dimensionless parameters (i.e., t-SNE dimension 1 and dimension 2).

(i.e., 100 kΩ), $R_{ADC}$ is the input impedance of ADC unit (i.e., 1 MΩ), and $R^i$ is the varying resistance of the $M_n$ sensor at $i$th channel. Afterward, the MCU was programmed with the customized codes to collect the multi-channeled voltage outputs in real time (as recorded in Supplementary Fig. 33), which were then sent out by the Bluetooth

chip. Our wireless sensor module demonstrated a data transmission rate ($S$) at ca. 450 bps (bits per second), which was defined in Eq. 6,

$$S = \frac{A}{t} \qquad (6)$$

where $A$ is the data bits of the Bluetooth chip and $t$ is the data transmission time. As we adopted a commercial Bluetooth HC-06, our wireless sensor module maintained its $S$ value above 400 bps in a broad space (>80 meters) and long-term (>100 h) continuous operation.

As shown in Fig. 4h, the wirelessly transmitted $M_n$ sensor data was input as the training data (Supplementary Table 2 in *GitHub*) to train a ML model based on ANN. Detailed ML framework is discussed in Supplementary Note 5. Figure 4i shows the t-distributed Stochastic Neighbor Embedding (t-SNE) scatterplot of multi-channeled $M_n$ sensor data in response to six full-body motions (see detailed description in Supplementary Note 6), and six distinct clusters were formed after dimension reduction. Without the use of image/video data, the ANN model was able to achieve a accuracy of 100% for full-body motion classification. The accuracy was defined in Eq. 7 and evaluated by using independent testing data (Supplementary Table 3 in *GitHub*),

$$Accuracy = 100\% - \frac{1}{N}\sum_{i=1}^{i=N}\frac{|P_i - T_i|}{T_i} \quad (7)$$

where $P^i$ is the determined type of $i$th full-body motion, and $T^i$ is the recorded label of $i$th full-body motion in testing data.

## Edge sensor module for in-sensor avatar reconstruction

Although the wireless sensor module ensured facile data connectivity to a terminal computing device, continuous transmission of multi-channeled sensor data led to high energy consumption for long-term monitoring[29,63]. In addition, the wireless transmission approach always faces interruption and disconnection challenges, especially when the wireless sensor modules are applied in distant fields or in underwater scenarios[64,65]. To address these challenges, edge data computing in a ML chip becomes an emerging approach to monitoring and determining full-body motions in high accuracy. By integrating seven $M_n$ sensors with a ML chip, an edge sensor module was fabricated with the equivalent circuit illustrated in Fig. 5a; the digital photo is shown in Supplementary Fig. 34. The edge sensor module was composed of seven $M_n$ sensors with joint-matched linear working windows, a multi-channeled ADC unit, a ML chip (with integrated Bluetooth function). It should be noted that the Bluetooth function was used to intermittently send the processed results out of the ML chip.

Our edge sensor module with in-sensor ML models was able to reconstruct personalized avatars that mimicked the continuous full-body motions in high precision and accuracy. Precise monitoring of continuous human motions has long been a primary center for various applications, such as gesture/gait recognition[26,27], animation production[66,67], remote healthcare[6,68], and virtual reality[9,11]. During the coronavirus pandemic, the technologies for sensing delicate body motions (e.g., trembling, shivering) become necessary, as the doctors can monitor the patients' symptoms in real time[1,69]. Also, the human motion sensing technologies have significant impacts in various industries, including sports[4,5], healthcare[69], and gaming entertainment[70,71].

In this work, the in-sensor ML model for avatar reconstruction was built up in two steps. First, the figure aspects and 15 joint locations of a volunteer were extracted from a pre-recorded video through an OPEN POSE program[72]. The detailed description of OPEN POSE program is provided in Supplementary Note 7. As shown in Fig. 5b, the stationary stickman avatar with 15 joints was constructed. Afterward, seven $M_n$ sensors with joint-match working windows were attached onto the joints of a volunteer, and the time-resolved $M_n$ sensor data were collected in a multi-channeled fashion, which were input as the training data (Supplementary Table 4 in *GitHub*) to train a ML model offline based on CNN (see details in Supplementary Note 8). The CNN model with optimal hyperparameters was then uploaded to activate the edge sensor module. When the volunteer performed a series of full-body

motions, the seven $M_n$ sensors were able to monitor the localized strain changes of different joints. As shown in Fig. 5c, the multi-channeled resistance change profiles were collected and transformed into the voltage outputs through ADC. By continuously receiving the voltage data, the ML chip with in-sensor CNN model enabled real-time and high-accuracy determination of 15 avatar joint locations, which were then transformed into personalized avatar animations by the OPEN POSE program (as shown in Supplementary Movie 4).

The determination error of in-sensor avatar reconstruction was calculated by measuring the difference between the real joint locations (extracted from camera-recorded videos, Supplementary Table 5 in *GitHub*) and the determined joint locations (computed by edge sensor module, Supplementary Table 6 in *GitHub*). The average determination error is defined in Eq. 8,

$$\text{Average Determination Error} = \frac{1}{15} \times \frac{170cm}{830} \times \sum_{i=0}^{i=14}\sum_{0}^{t}|P_t^i - T_t^i| \quad (8)$$

where $P_t^i$ is the CNN-determined location of $i^{th}$ joint at the time $t$, $T_t^i$ is the camera-recorded locations of $i^{th}$ joint at the time $t$, 170 cm is the volunteer's physical height, and 830 is the corresponding avatar height in the virtual coordinate system, and 15 is the number of monitored joints. By comparing 15 joint locations (Fig. 5d and Supplementary Fig. 35), the average avatar determination error was calculated to be 3.5 cm. As shown in Fig. 5e, the CNN-determined avatar animations successfully mimicked the volunteer's full-body motions in high precision and accuracy. It is worth mentioning that the avatar animation sometimes moved ahead of the full-body motions, specifically the squatting motions. As shown in Supplementary Fig. 36 and Supplementary Movie 4, the avatar's squatting movement (at the 22.6th second) was ahead of the video-recorded squatting motion (at the 23.0th second). The ahead motion determination was due to the early signals from the $M_p$ sensor on the back waist, where the preparatory muscular stretching happened before the squatting motions (see details in Supplementary Note 9). Supplementary Note 10 and 11 discuss more potentials of the edge sensor modules, including the improvement of the comfort levels of wearable $M_n$ sensor modules and the method to re-construct a 3D avatar (Fig. 5f and Supplementary Movie 5).

The edge sensor module demonstrated a critical advantage in power consumption over the wireless sensor module. Figure 5g compares the power consumption of wireless and edge sensor modules to reconstruct avatar animations (see detailed calculation in *Methods*). The wireless sensor module showed power consumption of 31.5 mW, the majority of which was used to wirelessly transmit multi-channeled sensor data to a terminal computing device (20 mW). It is worth noting that the power consumption of avatar reconstruction in the terminal computing device was neglected. On the other hand, the edge sensor module consumed 9 mW to achieve the same task of avatar reconstruction, which was 71% lower than the wireless one. The edge sensor module with high power efficiency is beneficial for full-body motion monitoring, as miniature batteries with limited energy capacities are normally used for wearable applications.

## Discussion

In this work, through harnessing the interfacial instability during localized thermal contraction, a variety of $M_n$ sensors with engineered MXene microtextures were fabricated, where the in-plane crack propagations and thus the strain sensing characteristics were systematically tuned (Fig. 6a). Four kinds of $M_n$ sensors with ultrahigh sensitivity (GF > 1000) and joint-matched working windows were attached onto the multi-joint surfaces of a volunteer to monitor full-body motions with high signal-to-noise ratios. As shown in Fig. 6b, by interfacing $M_n$ sensors with a Bluetooth chip, a wireless sensor module was assembled to achieve continuous and real-time streaming of multi-

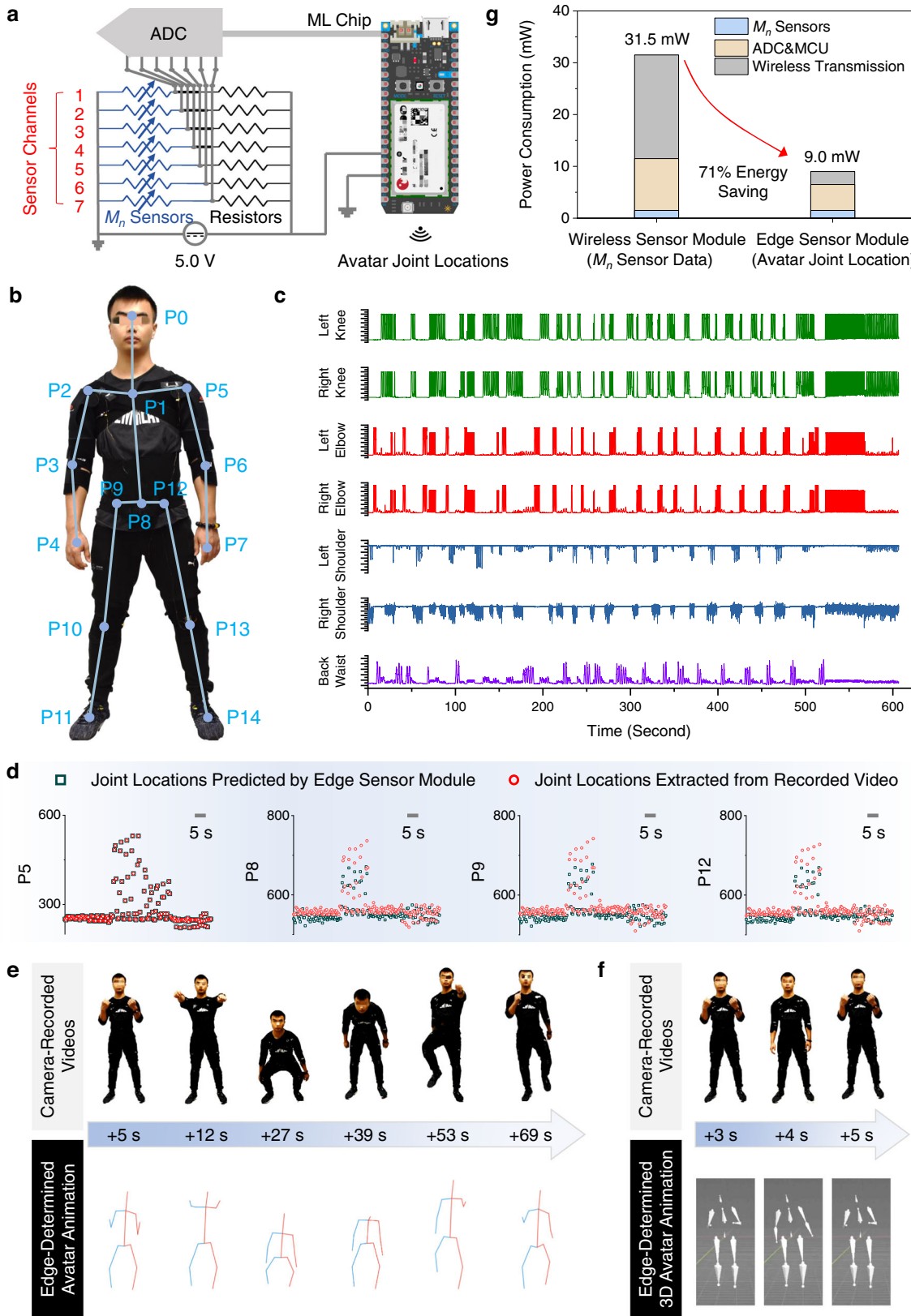

**Fig. 5 | Edge sensor module for in-sensor avatar reconstruction. a** Equivalent circuit of an edge sensor module. **b** Stickman avatar with 15 joints was customized based on a camera-recorded video. **c** Time-dependent $M_n$ sensor data in response to various full-body motions of a volunteer. **d** Joint location comparison between the determination results from an edge sensor module and the extracted information from a recorded video. **e** Comparison of full-body motions between the volunteer and the avatars constructed by the edge sensor module. **f** A 3D avatar was constructed by the edge sensor module. **g** Comparison of power consumption of wireless sensor module and edge sensor module toward avatar reconstruction.

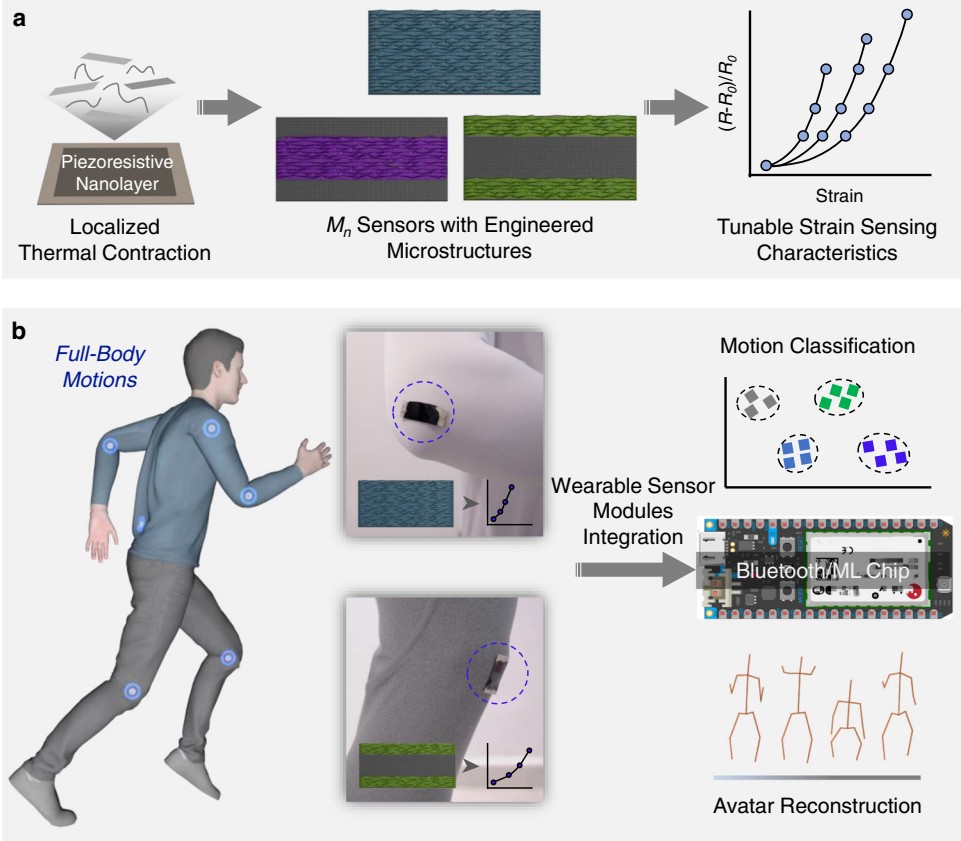

**Fig. 6 | Integration of wearable MXene sensors with edge computing for in-sensor avatar reconstruction. a** Through topographic design in the piezoresistive nanolayers by localized thermal contraction, the strain sensing characteristics of wearable MXene sensors can be programmably tuned to monitor delicate strain changes of different body joints. **b** Empowered by the integrated Bluetooth unit and ML chip, the strain sensor modules can be applied to achieve accurate full-body motion classification and avatar reconstruction.

channeled $M_n$ sensor data, and an ANN model was trained with 100% accuracy to classify various full-body motions. By further integrating $M_n$ sensors with a ML chip, an edge sensor module was fabricated for the in-sensor reconstruction of personalized avatar animations that mimic diverse full-body motions with an average avatar determination error of 3.5 cm, without external computing devices. Also, the standalone edge sensor module avoided wireless data streaming and demonstrated 71% less power consumption for avatar reconstruction than the wireless sensor module.

Our work has demonstrated multiple interdisciplinary advances. Our first advance is to develop a low-cost, scalable, and controllable approach to engineering wrinkle-like MXene microstructures via localized thermal contraction, which cannot be easily achieved by conventional bucking methods (Supplementary Figs. 37 and 38). Our second achievement is to control the crack propagation behaviors of piezoresistive nanolayers through topographic design including the $M_n$ nanolayers under different stretching directions, the $M_n$ nanolayers with varying the areal percentages of wrinkle-like region(s), and the $M_n$ nanolayers with varying positions and distribution of wrinkle-like region(s). As a result, the strain sensing characteristics of wearable strain sensor can be customized. Most of the reported works in the literature focuses on adjusting the piezoresistive nanolayer compositions to pursue high GFs, yet few strategies were developed to customize the linear working windows[23,30,35,47,62]. In this work, the linear working windows of our $M_n$ sensors can be tuned without sacrificing their ultrahigh strain sensitivity (GF > 1000). Third, compared with the state-of-the-art works in Table 1, our edge sensor module showed advances in terms of versatile integration method. As mentioned by multiple important review articles[29,69,73,74],

an urgent challenge in wearable sensors is to enable efficient transmission of large amounts of sensor data followed by in-sensor machine learning with power-efficient local computing. Multi-disciplinary efforts are required to address this challenge by advancing hardware/software development and optimizing the sensor/circuit interfaces. In this work, our edge sensor module was carefully designed to achieve full-body motion monitoring coupled with edge data processing and in-sensor machine learning, which has not been reported before. We believe that the integration method (i.e., wearable/stretchable sensors with edge computing chip(s)) demonstrated in this work is highly versatile and can benefit the advances in other fields, including wearable performance devices in sports and underwater soft robots.

## Methods
### Materials
Lithium fluoride (LiF, Sigma-Aldrich, BioUltra, ≥99.0%), hydrochloric acid (HCl, Sigma-Aldrich, ACS reagent, 37%), $Ti_3AlC_2$ MAX powders (MAX, Tongrun Info Technology Co. Ltd, China), single-walled carbon nanotubes (SWNTs, Timesnano Co. Ltd, China), sodium dodecyl sulfate (SDS, Sigma-Aldrich, >99.9%), poly(vinyl alcohol) (PVA) ($M_w$ 67,000, Sigma-Aldrich), ethanol (Thermo Fisher, >99.5%), and DCM (J.T. Baker; 99.9%) were used as received without further purification. Biaxial PS shrink films were purchased from Grafix. Thermally responsive shrink films (with uniaxial contraction mode) were produced by EVERGREEN SCIENTIICS. VHB™ Tape 4910 (1 inch × 36 yards) was purchased from 3 M. Silver paste was purchased from Ted Pella Inc. Commercial standard resistors were purchased from TELE-SKY. Deionized (DI) water (18.2 MΩ) was obtained from a Milli-Q water

**Table 1 | Comparison of this work with other wearable sensor systems for human motion monitoring**

| Reference | Sensor type | No. of sensor type (Location) | No. of sensors | Wireless data transmission | Signal acquisition | Motion classification | Avatar reconstruction | Edge computing |
|---|---|---|---|---|---|---|---|---|
| Jeong, H. et al. Proc. Natl. Acad. Sci. U. S. A. 2021. | Inertialsensor | 1(Arms, legs, head, chest) | 10 | Yes | Single-Channeled | × | ✓ | × |
| Tautges, J. et al. ACM Trans. Graph. 2011[19] | Inertialsensor | 1(Arms and legs) | 6 | Yes | Single-Channeled | × | ✓ | × |
| Luo, Y, et al. Nat. Electron. 2021[8]. | Strainsensor | 1 (All joints) | >200 | No | Multi-Channeled | ✓ | × | × |
| Wang, M. et al. Nat. Electron. 2020[26] | Strainsensor | 1(Hands) | 5 | No | Multi-Channeled | ✓ | × | × |
| Zhou, Z. et al. Nat. Electron. 2020[27]. | Strainsensor | 1(Hands) | 5 | Yes | Multi-Channeled | ✓ | × | × |
| Lin, M. et al. Adv. Mater. 2021[18] | Strainsensor | 1(Wrist and knees) | 4 | No | Single-Channeled | × | × | × |
| Kim, J. H. et al. Adv. Funct. Mater. 2021[2]. | Strainsensor | 1(Elbows and knees) | 4 | No | Single-Channeled | × | × | × |
| Yang, Z. et al. ACS Nano 2018[40] | Strainsensor | 1(All joints) | 8 | No | Single-Channeled | × | × | × |
| Sun, H. et al. ACS Appl. Mater. Interfaces 2019[79]. | Strainsensor | 1(Knees) | 2 | No | Single-Channeled | × | × | × |
| Ma, J. et al. ACS Appl. Mater. Interfaces 2019[80] | Strainsensor | 1(Elbows and knees) | 4 | No | Single-Channeled | × | × | × |
| Yang, H. et al. ACS Nano 2020[7]. | Strainsensor | 1(Hands) | 2 | Yes | Single-Channeled | × | × | × |
| This Work | Piezoresistive Sensor | 4(Back waist, shoulders elbows, knees) | 7 | Yes | Multi-Channeled | ✓ | ✓ | ✓ |

purification system (Merck Millipore) and used as the water source throughout the work.

### Preparation of $Ti_3C_2T_x$ MXene nanosheets

$Ti_3C_2T_x$ MXene nanosheets were prepared according to the literature[75]. 1.0 g of LiF was added to 6.0 M HCl solution (20 mL) under vigorous stirring. After the dissolution of LiF, 1.0 g of $Ti_3AlC_2$ MAX powder was slowly added into the HF-containing solution. The mixture was kept at 35 °C for 24 h. Afterward, the solid residue was washed with DI water several times until the pH value increased to ca. 7.0. Subsequently, the washed residue was added into 100 mL of DI water, ultrasonicated for 1 h, and centrifuged at 1308 g for 30 min. The supernatant was collected as the final suspension of $Ti_3C_2T_x$ MXene nanosheets with the concentration of ca. 5 mg mL$^{-1}$.

### Preparation of SWNT dispersion

The SWNT dispersion was obtained by adding the SWNT powders into the SDS solution (at the SDS concentration of 2 mg mL$^{-1}$) at the mass ratio of SWNT:SDS = 1:20. Then, the mixture was ultrasonicated for 2 h by a probe sonicator, and the concentration of the final SWNT dispersion was about 0.1 mg mL$^{-1}$.

### Calculation of dimension contraction of uniaxial shrink films

A uniaxial shrink film was cut into multiple rectangle-shaped pieces followed by thermal contraction in an oven at 100 °C. The dimensions of the uniaxial shrink film before and after thermal contraction were quantified by taking photographs and sampling their gray-scale line profiles using *ImageJ*. The dimension contraction of a uniaxial shrink film was calculated by Eq. 9,

$$\text{Dimension contraction}(-) = 1 - \frac{\text{Dimension after thermal contraction}}{\text{Dimension before thermal contraction}} \quad (9)$$

### Fabrication of freestanding $M_p$ nanolayers

Two dispersions of SWNTs and MXene nanosheets were mixed at different ratios, and 5 wt.% of PVA was then added into the SWNT−MXene dispersions. The as-prepared SWNT−MXene−PVA dispersions were next deposited onto PVDF membranes (0.22 μm pore, Merck Millipore) through vacuum-assisted filtration systems. To remove SDS residues, the filtered SWNT−MXene−PVA thin films (abbreviated as ps-MXene nanolayers) were rinsed with excessive DI water. Afterward, freestanding ps-MXene nanolayers were detached from the PVDF membranes by immersing them in ethanol. The planar ps-MXene nanolayers were categorized and abbreviated as $M_p$ nanolayers.

### Fabrication of freestanding $M_w$ nanolayers

A uniaxial shrink film was cut into multiple rectangle-shaped pieces (4 × 8 cm$^2$), washed with ethanol, and dried under N$_2$ flow. The cut shrink films were next treated with oxygen plasma for 2 min to enhance the hydrophilic interactions between PS substrates and $M_p$ nanolayers[76]. Afterward, the $M_p$ nanolayers were carefully transferred onto the plasma-treated uniaxial shrink films followed by overnight drying. The $M_p$ nanolayer-coated shrink films were then heated in an oven at 100 °C for 120 s to induce uniaxial thermal contraction. By harnessing surface instability during thermal contraction, the $M_p$ nanolayers were deformed into wrinkle-like microtextures (abbreviated as $M_w$ nanolayers). The shrunk samples were then immersed in DCM to dissolve the PS substrates to obtain freestanding $M_w$ nanolayers (see SEM image in Fig. 1b), which were sequentially rinsed with DCM, acetone, and ethanol. The $M_w$ nanolayers were stored in ethanol.

## Fabrication of freestanding M$_{p\text{-}w\text{-}p}$ nanolayers

A uniaxial shrink film was cut into multiple rectangle-shaped pieces ($4 \times 8$ cm$^2$), washed with ethanol, and dried under N$_2$ flow. The cut shrink films were next treated with oxygen plasma for 2 min to enhance the hydrophilic interactions between PS substrates and $M_p$ nanolayers. Afterward, the $M_p$ nanolayers were carefully transferred onto the plasma-treated shrink films followed by overnight drying. Then, the two ends of the uniaxial shrink films were fixed by adhering thin glasses at their backside (see schematic illustration in Fig. 1a). The $M_p$ nanolayer-coated shrink films were then heated in an oven at 100 °C for 120 s to induce uniaxial thermal contraction. By harnessing surface instability during thermal contraction, the $M_p$ nanolayers were deformed into the $M_n$ nanolayers with wrinkle-like microtextures localized only in the middle region (abbreviated as $M_{p\text{-}w\text{-}p}$ nanolayer). The shrunk samples were then immersed in DCM to dissolve the PS substrates to obtain freestanding $M_{p\text{-}w\text{-}p}$ nanolayers (see SEM image in Fig. 1b). It is worth to note that the $M_{p\text{-}w\text{-}p}$ nanolayers were attached on PS shrink films, and the glass slides were adhered at the backside of shrink film. In other words, the textured $M_{p\text{-}w\text{-}p}$ nanolayer and the glass slides were attached to the two sides of a shrink film (see Supplementary Fig. 10). After the $M_{p\text{-}w\text{-}p}$-coated PS samples were immersed in a DCM bath, the middle PS shrink film was dissolved, and the $M_{p\text{-}w\text{-}p}$ nanolayers were detached and became freestanding. The freestanding $M_{p\text{-}w\text{-}p}$ nanolayers were sequentially rinsed with DCM, acetone, and ethanol. The $M_{p\text{-}w\text{-}p}$ nanolayers were stored in ethanol.

## Fabrication of freestanding M$_{w\text{-}p\text{-}w}$ nanolayers

A uniaxial shrink film was cut into multiple rectangle-shaped pieces ($4 \times 8$ cm$^2$), washed with ethanol, and dried under N$_2$ flow. The cut shrink films were next treated with oxygen plasma for 2 min to enhance the hydrophilic interactions between PS substrates and $M_p$ nanolayers. Afterward, the $M_p$ nanolayers were carefully transferred onto the plasma-treated shrink films followed by overnight drying. Then, the middle sections of the uniaxial shrink films were fixed by adhering thin glasses at their backside (see schematic illustration in Fig. 1a). The ps-MXene-coated PS device was then heated in an oven at 100 °C for 120 s to induce uniaxial thermal contraction. By harnessing surface instability during thermal contraction, the $M_p$ nanolayer was deformed into the $M_n$ nanolayers with wrinkle-like microtextures localized only in the edge regions (abbreviated as $M_{w\text{-}p\text{-}w}$ nanolayer). The shrunk samples were then immersed in DCM to dissolve the PS substrates to obtain freestanding $M_{p\text{-}w\text{-}p}$ nanolayers (see SEM image in Fig. 1b). It is worth to note that the $M_{w\text{-}p\text{-}w}$ nanolayers were attached on PS shrink films, and the glass slides were adhered at the backside of shrink film. In other words, the textured $M_{w\text{-}p\text{-}w}$ nanolayer and the glass slides were attached to the two sides of a shrink film (see Supplementary Fig. 10). After the $M_{w\text{-}p\text{-}w}$-coated PS samples were immersed in a DCM bath, the middle PS shrink film was dissolved, and the $M_{p\text{-}w\text{-}p}$ nanolayers were detached and became freestanding. The freestanding $M_{w\text{-}p\text{-}w}$ nanolayers were sequentially rinsed with DCM, acetone, and ethanol. The $M_{w\text{-}p\text{-}w}$ nanolayers were stored in ethanol.

## Finite element analysis (FEA) simulation

The 3D models of $M_p$, $M_w$, $M_{p\text{-}w\text{-}p}$, and $M_{w\text{-}p\text{-}w}$ microstructures were built by using SolidWorks 2018, and their surface characteristics were modelled by the Freeform feature. The FEA models of $M_p$, $M_w$, $M_{p\text{-}w\text{-}p}$, and $M_{w\text{-}p\text{-}w}$ nanolayers were further built by the Static Structural module of ANSYS Workbench 19.0 (see Supplementary Fig. 39). The simulation parameters for ps-MXene nanolayers were set as follows: Young's modulus -1.7 GPa, Poisson's ratio 0.227, and mass density 1.25 g cm$^{-3}$. Cartesian coordinate was chosen for the mesh method, and the element size was set to be 100 μm. The FEA simulation was conducted as shown in Supplementary Fig. 40, where the left boundary of the 3D model was fixed, while the right boundary was set to be movable along $x$ and $y$ directions. Uniaxial stretching was simulated by moving the right boundary, and the equivalent elastic strains and the overall deformation were recorded.

## Fabrication of M$_p$, M$_w$, M$_{p\text{-}w\text{-}p}$, and M$_{w\text{-}p\text{-}w}$ strain sensors (M$_n$ sensors)

The freestanding $M_p$, $M_w$, $M_{p\text{-}w\text{-}p}$, and $M_{w\text{-}p\text{-}w}$ nanolayers in ethanol were carefully transferred onto VHB™ tapes followed by overnight drying. Copper wires were connected to the two ends of the $M_n$ nanolayers, and silver paste was applied between ps-MXene nanolayers and copper wires to ensure good electrical contacts. The resistance profiles of $M_p$, $M_w$, $M_{p\text{-}w\text{-}p}$, and $M_{w\text{-}p\text{-}w}$ strain sensors were monitored by Industrial Multimeters (EX503).

## Calculation of crack-to-width ratios ($\varphi$) and crack densities ($\rho$) of M$_n$ sensors

The crack-to-width ratios ($\varphi$) and crack densities ($\rho$) of $M_n$ sensor were investigated *via* in situ SEM. The definitions of $\varphi$ and $\rho$ are described in Eqs. 10 and 11, respectively,

$$\varphi = \frac{\max(L_{crack})}{W} \tag{10}$$

$$\rho = \frac{\sum(L_{crack})}{A} \tag{11}$$

where $\max(L_{crack})$ is the length of the longest surface crack, $\sum(L_{crack})$ is the cumulative length of all surface cracks, and $A$ and $W$ are the area and width of a $M_n$ nanolayer, respectively.

## Circuit design of a wireless sensor module

The equivalent circuit of the wireless sensor module is described in Fig. 4g. The wireless sensor module was designed with seven data collection channels, which consisted of seven $M_n$ sensors (including one $M_p$, two $M_{p\text{-}w\text{-}p}$, two $M_w$, and two $M_{w\text{-}p\text{-}w}$ sensors), an ADC (AD7606, Risym), a MCU (MCU-PCA9658, DeXin Electronics), and a Bluetooth module (HC-06, XinTaiWei Electronics). Each $M_n$ sensor was connected to a standard resistor (100 kΩ) in series. By applying a constant voltage of 5.0 V, the ADC unit measured the voltage outputs ($V_s^i$) in real time, which were derived in Eq. 5. Afterward, the multi-channeled voltage signals were processed by MCU, and sent out by Bluetooth,

## Circuit optimization of a wireless sensor module

First, to achieve high sensitivity of wireless sensor module, a high-resolution 16-bit ADC (AD7606, Risym) was adopted. Second, standard resistors were used as the regulator to enable high measurement accuracy with low wireless transmission errors, which were calculated by following Eq. 12,

$$Wireless\ Transmission\ Error = \frac{|R_{input} - R_{output}|}{R_{input}} \tag{12}$$

where $R_{input}$ is the resistance value of a $M_n$ sensor measured directly by a digital multimeter, and $R_{output}$ is the output resistance value of a $M_n$ sensor wirelessly transmitted and converted from the ADC-recorded voltage value.

In this work, we measured wireless transmission error by using the $M_n$ sensors under different strains. As shown in Supplementary Fig. 32, with 100 kΩ standard resistors, the wireless sensor module exhibited low and strain-stable transmission errors <4%. In the other words, a $M_{w\text{-}p\text{-}w}$ sensor under 48% strain (with a 39.1 kΩ resistance) was read as 37.7 kΩ after wireless data transmission, showing a transmission error of 3.6%. On the other hand, with 5 kΩ standard resistors, the wireless sensor module exhibited higher and strain-dependent transmission errors (the average error >20%). For instance, a $M_{w\text{-}p\text{-}w}$ sensor under

48% strain (with a 39.1 kΩ resistance) was read as 13.3 kΩ after wireless data transmission, showing a transmission error of 65%.

On the other hand, to simultaneously record the signals of multiple $M_n$ sensors, the execute codes were programmed into MCU (the codes were provided in *GitHub*: https://github.com/jiali1025/Wearable-MXene-Sensors-with-In-Sensor-Machine-Learning-for-Full-Body-Avatar-Reconstruction).

## Power consumption of a wireless sensor module

The power consumption ($P$) of the wireless sensor module is calculated based on Eqs. 13 and 14

$$\frac{1}{R_{module}} = \frac{1}{R_{M_p} + R_s} + \frac{2}{R_{M_w} + R_s} + \frac{2}{R_{M_{p-w-p}} + R_s} + \frac{2}{R_{M_{w-p-w}} + R_s} \quad (13)$$

$$P = \frac{V^2}{R_{module}} + P_{ADC} + P_{MCU} + P_{BLE} \quad (14)$$

where $R_s$ is the resistance of the standard resistor (i.e., 100 kΩ), and $R_{M_p}$, $R_{M_{p-w-p}}$, $R_{M_w}$, and $R_{M_{w-p-w}}$ are the resistances of $M_p$, $M_{p-w-p}$, $M_w$, and $M_{w-p-w}$ sensors, respectively; $R_{module}$ is the resistance of the wireless sensor module, $P_{ADC}$, $P_{MCU}$, and $P_{BLE}$ are the power consumptions of ADC (ca. 5 mW), MCU (ca. 5 mW), and Bluetooth units (ca. 20 mW), respectively. $V$ is the applied voltage (i.e., 5.0 V).

## Circuit design of an edge sensor module

As shown in Supplementary Fig. 34, seven $M_n$ sensors were integrated with an ADC unit and a ML chip (ARDUINO, Element14 Pte Ltd) in series. The equivalent circuit is illustrated in Fig. 5a. Basically, the ADC unit collected multi-channeled $M_n$ sensor data and send them to the ML chip. Then, the ML model (which was trained offline) was able to process the $M_n$ sensor data followed by the determination of full-body avatar joint locations. The generated avatar joint locations were then transmitted by using the Bluetooth chip integrated in the ML chip. The execute codes were programmed into the ML chip (the codes were provided in *GitHub*: https://github.com/jiali1025/Wearable-MXene-Sensors-with-In-Sensor-Machine-Learning-for-Full-Body-Avatar-Reconstruction).

## Power consumption of an edge sensor module

The power consumption ($P$) of the edge sensor module is calculated based on Eqs. 15 and 16

$$P = \frac{V}{R_{module}} + P_{ADC} + P_{MLChip} \quad (15)$$

$$P_{MLChip} = V \cdot I_{MLChip} \cdot \frac{1477}{60000} \quad (16)$$

where $R_{module}$ is the resistance of the edge sensor module calculated by Eq. 13, $P_{ADC}$ is the power values of ADC (ca. 5 mW), and $V$ is the applied voltage (i.e., 5 V), $I_{Chip}$ is the current required for the ML chip (the average current dissipation is 21.5 mA). For the ML chip, it required 1,477 regressions for 1 min avatar motion reconstruction, and each regression took about 1 ms.

## Characterization

XRD analyses were conducted using an X-ray diffractometer (Bruker, D8 Advance X-ray Powder Diffractometer, Cu Kα ($\lambda = 0.154$ radiation)) at a scan rate of 4° min$^{-1}$. The as-prepared MXene nanosheets were characterized by using a high-resolution transmission electron microscopy (HRTEM, JEOL 2010F). The surface morphologies of $M_p$, $M_w$, $M_{p-w-p}$, and $M_{w-p-w}$ nanolayers were characterized by using a SEM (FEI Quanta 600) and a field emission SEM (JEOL-JSM-6610LV)

operating at 15.0 kV. Surface roughness of $M_p$, $M_w$, $M_{p-w-p}$, and $M_{w-p-w}$ nanolayers was measured by AFM (Model: Bruker Dimension ICON). The characteristic crack lengths and the crack densities of $M_p$, $M_w$, $M_{p-w-p}$, and $M_{w-p-w}$ nanolayers were quantified by using *ImageJ*. Fatigue tests were performed on the $M_w$ strain sensor for 2000 cycles under repeated uniaxial strains, which were performed on a tensile tester (Instron 5543, Instron, USA) with a 500 N load cell.

## Data availability

The data generated in this study are provided in the Supplementary Information/Source Data file. Supporting files of Supplementary Tables 2–6 generated in this study have been deposited in the public *GitHub* (https://github.com/Haitao008/Supporting-Tables) and Zenodo[77] (https://doi.org/10.5281/zenodo.7012006) without any restrictions. Source data are provided with this paper.

## Code availability

The Python code to implement the machine learning tasks in this study have been deposited in the public *GitHub* (https://github.com/jiali1025/Wearable-MXene-Sensors-with-In-Sensor-Machine-Learning-for-Full-Body-Avatar-Reconstruction) and Zenodo[77] (https://doi.org/10.5281/zenodo.7012006) without any restrictions.

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

## Acknowledgements

We thank Dr. Lei Zhang (Zhongbei University) and Mr. Zhiyao Luo (Oxford University) for discussions about the circuit design and machine learning frameworks. The authors acknowledge the financial support provided by the Start-Up Fund of University of Maryland, College Park (KFS No.: 2957431 to P.-Y. C.), the MOST-AFOSR Taiwan Topological and Nanostructured Materials Grant under Grant No. FA2386-21-1-4065 (KFS No.: 5284212 to P.-Y. C.), and the Maryland Energy Innovation Institute (MEI²) Energy Seed Grant (KFS No.: 2957597 to P.-Y. C.).

## Author contributions

P.-Y.C. and H.Y. conceived the project ideas and designed the experiments. H.Y. carried out most wet-lab based experiments including the synthesis of MXene nanosheets, sensor modules' fabrication and characterizations. X.X. and H.Y. performed the FEA simulations and analyses. H.Y., J.L., and Y.L. designed the machine learning framework. J.W. and H.Y. designed the wireless sensor module. H.Y., J.L., J.W., and Y.L. collected the multi-channeled sensor data. S.S. designed the edge sensor module and calculated the power consumption. P.-Y.C. and H.Y. interpreted the results and co-wrote the paper. K.L., Z.L., H.C.Y., Q.W., J.Y., and P.-L.Y. involved in the discussion and paper revisions. P.-Y.C, S.S., X.W., J.S.H., and K.M. supervised this project.

## Competing interests

The authors declare no competing interests.
