## [Peer Review File · Nature Communications]

REVIEWER COMMENTS

Reviewer #1 (Remarks to the Author):

The paper reported a MXene/CNTs/PVA composite film strain sensor with both adjustable sensing range (from 6% to 84%) and high sensitivity (over 1000). The materials used in this work are common, but their preparation methods are interesting. In this work, the MXene composite film forms a controllable wrinkle-like topographies by pasting on the heat-shrinkable PS film, and this folded structure has a significant impact on the sensing performance. This is an interesting method to regulate the structure of flexible strain sensors. After that, the paper devoted a lot of space to the construction of the material on the integrated device, including the construction of wireless integrated devices that facilitate signal transmission and wired integrated devices that can reduce energy consumption, but I am not sure whether the construction of this integrated device is novel enough. Major issues are required to address to further improve this work.

1. In figure S2, the authors wants to show " SWNTs were well dispersed within the ps-MXene nanolayer " through the SEM section of the MXene composite film, but the picture does not reflect the meaning of the author due to the shielding of a large number of CNTs. The author should replace this picture to better illustrate the problem.

2. The scale bar marked in figure S7b is wrong.

3. In this paper, the stretchability of the device is enhanced by forming "wrinkle-like topographies" on the surface. In fact, many works have reported on this way, in which tensile direction is perpendicular to the fold direction, and the tensile enhancement of the device under this tensile direction is easy to understand. However, the stretching direction in this work is parallel to the folding direction. Why should the author choose this stretching direction? What is the difference or advantage between this direction and other conventional stretching directions?

4. In Figure 2a, the author explains why folds parallel to the tensile direction can enhance the tensile capacity of the device through finite element simulation. However, given this phenomenon that is not commonly seen, the author should describe the principle of this phenomenon in details.

5. The research on the wrinkles phenomenon in this paper focuses on G0 without wrinkles, G1F covering wrinkles, G1E of edge wrinkles, and G1M of the middle wrinkles, but the distribution of wrinkles is far more than these four kinds. Compared with this kind of research, the readers may be

more interested in the proportion of the distribution area of wrinkles to the total area and the influence of the distribution position of wrinkles on the sensing performance of devices.

6. In Fig. 3a, it can be seen that the tensile property of G1F with wrinkles on the surface is greater than G1M with wrinkles in the middle but less than G1E with wrinkles on the edge. The authors should explain this phenomenon.

7. As for the naming method in this article, write "the planar ps-MXene nanolayers (named as G0) " in line 1 on page 6, but the "G" after the abbreviation is not related to the original word, which may cause inconvenience to readers. Authors should name abbreviations that are easier to understand.

8. In Figure 3d, the "signal errors" value of G0 is much larger than the other three sensors, which is interpreted by the author as " where large surface cracks were randomly generated under strains.". I think this may be caused by the poor dispersion of nanomaterials. According to the author's explanation, the "signal errors" value of G1F should be much higher than that of the other three materials because of its most folded distribution.

9. In Figure S17, the G1F sensor has worked continuously for 2000 times under 30% strain, but this demonstrated lifetime is far from enough. Human joints typically moved thousands of times a day. I think the author should cycle test all types of sensors used for integration to determine their application potential.

10. Was the integration process for the sensing devices used in this work an innovative integration method? Or just a universal method?

Reviewer #2 (Remarks to the Author):

In the manuscript, wearable Ti₃C₂T_x MXene sensor modules are fabricated with in-sensor machine learning (ML) models, either functioning via wireless streaming or edge computing, for full-body motion classifications and avatar reconstruction. Through topographic design on piezoresistive nanolayers, the wearable strain sensor modules exhibited ultrahigh sensitivities within the working windows and meet all joint deformation ranges. By integrating the wearable sensors with a ML chip,

an edge sensor module is fabricated, enabling in-sensor reconstruction of high-precision avatar animations that mimic continuous full-body motions with 92.6% accuracy. However, the following problems and issues should be addressed.

1. In the manuscript, a lot of mechanical and electrical characterizations of the wrinkle-like MXene textures should be provided.
2. In the manuscript, the cyclic stability of the sensor should be considered.
3. In the application part of the sensors, the significance of mimicking continuous full-body motions should be provided?
4. In Fig.3a, why the relative resistance is changed by the thickness of Gn nanolayers?
5. In the manuscript, the innovation and mimic continuous full-body motions should be provided clearly.

Reviewer #3 (Remarks to the Author):

In this manuscript, the Gn sensors were fabricated with Ti₃C₂T_x MXene, SWNTs and PVA. Through topographic design during localized thermal contraction, the Gn sensor modules exhibited ultrahigh sensitivities within the working windows that met various joint deformation ranges. Neural networks were used to fuse the sensor data for full-body motion classifications and avatar reconstruction. Edge sensor module with ML chip was used to reduce wireless data flow and power consumption.

The work is interesting, but the following questions need to be concerned and addressed before considering the publication:

1. The strain sensor needs to be further tested to evaluate its performance, such as hysteresis, response time. Please provide enlarged profiles of step responses under dynamic stretching and releasing (> 30% strain). Peak shoulders appear when the strain >25% as shown in Fig. 3b. Please clarify.
2. The reviewer suggests to replace “prediction” by “estimation” or “determination” in the manuscript, as “prediction” generally refers to predictive estimation ahead of time. If so, the leading time should be given?

3. The authors mentioned “By continuously receiving the voltage data, the ML chip with in-sensor CNN model enabled real-time and high-accuracy prediction of 15 avatar joint locations.”

How to determine the fixed time length for the grid-like input data of CNN? The time delay would be serious if the time length was long.

The proposed method for evaluating the estimation accuracy of joint locations (i.e. eq. 8) is incorrect and unreasonable. The authors proposed a relative index by dividing location errors by absolute location T_i . The absolute location T_i is self-defined coordinate position, cannot be as the denominator. The accuracy would be tuned to be very high if setting a large value of T_i . The accuracy should be evaluated using absolute location error with physical unit (e.g. mm or cm) rather than relative expression.

4. The present work only concerned two-dimensional reconstruction using camera recorded video as reference data. However, human body/limb motions are actually three-dimensional movements. How to solve this problem?

5. In Figure 2c, it seems that the marking of texture region and planar region of G1-E is not correct? Please check.

6. What are the resistance value and electric conductivity of G_n sensors under non-tensile strain?

7. In Page 10, the authors mentioned “Our wireless sensor module demonstrated a high data transmission rate (S) at ca. 450 bps”. This transmission rate is very slow and cannot be expressed as “high”. Usually, the transmission rate of Bluetooth can reach about 1 Mbps.

In Page 11, the authors also mentioned “Our wireless sensor module maintained its high S values above 400 bps within 80 meters (Fig. S24a), across 4 cement brick walls (one wall thickness ~ 20 cm) (Fig. S24b), and after 100-hour continuous operation (Fig. S25).” This performance is attributed to the commercial Bluetooth HC-06, not the contribution of the proposed sensor module. The reviewer suggests to modify the description and delete Fig. S24.

8. There are problematic issues in the statement of “Each G_n sensor was connected in series with standard resistors, and the resistor value was specifically selected to be $100\text{ k}\Omega$ to ensure wireless data transmission with high accuracy.”, Fig. 4g, Fig. S22 and Methods “Circuit optimization of a wireless sensor module.”

The mentioned “transmission error” is not caused by the standard resistor value, it is due to the inherent input resistor of ADC and can be eliminated by adding the voltage follower. Some definitions are unclear. For example, the “Feed-in resistance” in Fig S22 requires the reader to guess what resistance it refers to, in the statement “where R_{input} is the resistance of the standard resistors (from 0.1 to 100 k Ω) used to replace G_n sensors in the wireless sensor module. The wireless sensor module with 100-k Ω standard resistors showed low transmission errors of 3%,” the two “standard resistors” do not actually refer to the same resistance, which may confuse readers.

It is suggested that the authors should improve the circuit design and clarify the confusion of the circuit description.

9. The authors mentioned: “The freestanding nanolayers in ethanol were carefully transferred onto VHB tapes followed by overnight drying.”

According to our knowledge, the rebound performance of VHB tape is very poor and the hysteresis is very large. How to ensure the measurement accuracy of the sensor during movement using VHB tape? The reviewer suggests to provide the strain response profiles of the sensor under dynamic stretching and releasing for several cycles? Hysteresis of the sensor needs to be tested and evaluated.

10. The proposed devices still have the limitations of inconvenient/uncomfortable wearing, wires spreading all over human body and limbs, which seriously affects the user's movement experience.

11. How to adhere the MXene nanolayer onto the glass shown in Fig. 1a, and how to separate the MXene nanolayer from the glass after the fabrication?

12. In movie S4, the avatar animations are sometimes ahead of the body motion, sometimes lag behind. Why? Avatar movement ahead of time is impossible and unreasonable.

Responses to the Reviewers' Comments

Reviewer #1

Comment 1: *The paper reported a MXene/CNTs/PVA composite film strain sensor with both adjustable sensing range (from 6% to 84%) and high sensitivity (over 1000). The materials used in this work are common, but their preparation methods are interesting. In this work, the MXene composite film forms a controllable wrinkle-like topographies by pasting on the heat-shrinkable PS film, and this folded structure has a significant impact on the sensing performance. This is an interesting method to regulate the structure of flexible strain sensors. After that, the paper devoted a lot of space to the construction of the material on the integrated device, including the construction of wireless integrated devices that facilitate signal transmission and wired integrated devices that can reduce energy consumption, but I am not sure whether the construction of this integrated device is novel enough. Major issues are required to address to further improve this work.*

Response 1: We thank Reviewer #1 for his/her careful review and valuable comments. We have made point-by-point responses for each comment below. Regarding Reviewer #1's concern about the novelty of our integrated wireless and edge sensor modules, we have provided a detailed discussion in **Response 11**.

Comment 2: *In figure S2, the authors wants to show " SWNTs were well dispersed within the ps-MXene nanolayer " through the SEM section of the MXene composite film, but the picture does not reflect the meaning of the author due to the shielding of a large number of CNTs. The author should replace this picture to better illustrate the problem.*

Response 2: We appreciate Reviewer #1's careful review. Per Reviewer #1's request, we have taken new SEM images in **Fig. S2b** and **S2c**, which show that SWNTs were highly dispersed and entangled within the MXene multilayer.

Fig. S2 Characterizations of ps-MXene nanolayers. (a) Top-down SEM image of a ps-MXene nanolayer. The mass ratio of MXene/SWNT/PVA (wt.%) was set at 85/10/5. Interconnected SWNTs were observed on the surface of ps-MXene nanolayer. (b)(c) Cross-sectional SEM images of a ps-MXene nanolayer. SWNTs were well dispersed within the ps-MXene nanolayer. (d) XRD patterns of $Ti_3C_2T_x$ MXene, SWNTs, and ps-MXene nanolayer. The ps-MXene nanolayer exhibited the (002) diffraction peak of MXene multilayers at 6.04° and the representative peaks of SWNTs at 24.5° , indicating the prospective assembly of MXene nanosheets and SWNTs. (e) Raman spectra of MXene nanosheets, SWNTs, and ps-MXene nanolayer (at the MXene/SWNT/PVA ratio of 85/10/5).

Revision made

Page S6 of Supporting Information.

Comment 3: *The scale bar marked in figure S7b is wrong.*

Response 3: We apologize for the mistake and appreciate Reviewer #1's careful review. We have corrected the scale bar in **Fig. S7b**.

Fig. S7 (a) SEM image and (b) high-resolution SEM image of a M_n nanolayer.

Revision made

Page S11 of Supporting Information.

Comment 4: *In this paper, the stretchability of the device is enhanced by forming “wrinkle-like topographies” on the surface. In fact, many works have reported on this way, in which tensile direction is perpendicular to the fold direction, and the tensile enhancement of the device under this tensile direction is easy to understand. However, the stretching direction in this work is parallel to the folding direction. Why should the author choose this stretching direction? What is the difference or advantage between this direction and other conventional stretching directions?*

Response 4: There are two major advantages of selecting parallel stretching over the perpendicular stretching, including (1) higher M_n sensors' sensitivities and wider linear

working windows and **(2)** more design opportunities *via* topographic design. As shown in **Fig. S41**, the M_n sensors (including M_w , M_{w-p-w} , M_{p-w-p}) under parallel stretching demonstrated wider linear working windows (with an average strain range of 15%) and higher sensitivities (GF >1,100) than the M_n sensors under perpendicular stretching (an average strain range of 6%, GF ~600).

To investigate the effect of stretching directions (perpendicular, 45°, and parallel) on the crack propagation behaviors of all M_n sensors (from M_w to M_{p-w-p} and M_{w-p-w}), additional FEA simulation and *in situ* SEM have been conducted in the revised manuscript.

First, for the M_w sensor, **Fig. S42a** and **S42b** show the FEA and *in situ* SEM results under perpendicular and parallel stretching. When stretched perpendicularly, the M_w nanolayer experienced attenuated strains (<10%, quantified by *ImageJ*), and the periodic wrinkles were first unfolded into a planar morphology and then quickly formed large, long cracks (**Fig. S42a**). Thus, the M_w sensor under perpendicular stretching demonstrated a relatively lower sensitivity (GF < 400) and a narrower linear working window of 37–43% (**Fig. S42c**). In comparison, when stretched parallelly, the M_w nanolayer experienced moderate strains (ca. 20%), and short/zigzag cracks propagated accordingly (**Fig. S42b**). The M_w sensor under perpendicular stretching demonstrated a higher GF of 1,230 and a wider linear working window of 25%–39% (**Fig. S42c**).

Second, for the M_{p-w-p} sensor, **Fig. S43a** and **S43b** show the FEA and *in situ* SEM results under perpendicular and parallel stretching. When the M_{p-w-p} nanolayer was stretched perpendicularly, the side planar regions experienced large strains (>50%), and long cracks emerged and quickly cut off the conductive pathways (**Fig. S43a**). As a result, the M_{p-w-p} sensor under perpendicular stretching demonstrated a relatively lower sensitivity (GF < 700) and a narrower linear working window of 11–17% (**Fig. S43c**). In comparison, when the M_{p-w-p} nanolayer was stretched parallelly, the middle region (with wrinkle-like textures) experienced attenuated strains (<20%) and developed short/zigzag cracks. Although the side planar regions showed long and continuous cracks, the middle region prevented the conductive pathways from being completely cut off (**Fig. S43b**). As a result, the M_{p-w-p} sensor under perpendicular stretching showed a higher GF of 1,160 and a wider linear working window of 8–24% (**Fig. S43c**).

Third, for the M_{w-p-w} sensor, **Fig. S44a** and **S44b** show the FEA and *in situ* SEM results under perpendicular and parallel stretching. The M_{w-p-w} sensor demonstrated similar crack propagation behaviors as the M_{p-w-p} sensor. When the M_{w-p-w} nanolayer was stretched perpendicularly, the middle planar region developed long cracks that quickly cut off the conductive pathways (**Fig. S44a**). As a result, the M_{w-p-w} sensor under perpendicular stretching demonstrated a narrower linear working window of 12–18% (**Fig. S44c**). In comparison, when stretched parallelly, the M_{w-p-w} nanolayer experienced attenuated strains (<20%) in the edge regions (with wrinkle-like textures), where short/zigzag cracks emerged. Although the middle planar region showed long and continuous cracks, the edge regions prevented the conductive pathways from being completely cut off (**Fig. S44b**). As a result, the linear working window of a M_{w-p-w} sensor expanded to 35–50% by changing the stretching direction (**Fig. S44c**).

Futhermore, for the M_{w-p-w} sensor, additional studies were carried out under 45° stretching in **Fig. S45**. When the M_{w-p-w} nanolayer was under 45° stretching, the strain was divided into two directions: one is parallel, and the other is perpendicular to the wrinkle axes. The M_{w-p-w} nanolayer under 45° stretching demonstrated the FEA results with mixed effects. According to the results from **Fig. S42 to S44**, the M_n sensors under parallel stretching normally demonstrated wider working windows, and the M_n sensors under perpendicular stretching showed lower sensitivities (i.e., GF). As a result, as shown in **Fig. S45**, the M_{w-p-w} sensor under 45° stretching showed a moderate working window of 21–30% yet a low GF of 484.

Considering the advantages of higher strain sensitivities and wider linear working windows, we adopted the parallel stretching direction for all M_n sensors. We have added the above results and discussions in the revised manuscript to make this part clearer.

Fig. S41 M_n sensors' performance under different stretching directions. (a) The stretching direction is parallel to the wrinkle axes. **(b)** The stretching direction is perpendicular to the wrinkle axes.

Fig. S42 M_w sensor performance under different stretching directions. (a) FEA simulation and *in situ* SEM image of M_w nanolayer under perpendicular stretching. (b) FEA simulation and *in situ* SEM image of M_w nanolayer under parallel stretching. (c) Strain sensing curves of M_w sensor under perpendicular and parallel stretching.

Fig. S43 M_{p-w-p} sensor performance under different stretching directions. (a) FEA simulation and *in situ* SEM image of M_{p-w-p} nanolayer under perpendicular stretching. (b) FEA simulation and *in situ* SEM image of M_{p-w-p} nanolayer under parallel stretching. (c) Strain sensing curves of M_{p-w-p} sensor under perpendicular and parallel stretching.

Fig. S44 M_{w-p-w} sensor performance under different stretching directions. (a) FEA simulation and *in situ* SEM image of M_{w-p-w} nanolayer under perpendicular stretching. (b) FEA simulation and *in situ* SEM image of M_{w-p-w} nanolayer under parallel stretching. (c) Strain sensing curves of M_{w-p-w} sensor under perpendicular and parallel stretching.

Fig. S45 M_{w-p-w} sensor performance under different stretching directions. (a) Strain sensing curves of M_{w-p-w} sensor under perpendicular, 45°, and parallel stretching. (b) FEA simulation of M_{w-p-w} nanolayer under perpendicular, 45°, and parallel stretching.

Revision made

Page 8, 27 of revised manuscript and Page S45–S49, S61–S63 of Supporting Information.

Comment 5: In Figure 2a, the author explains why folds parallel to the tensile direction can enhance the tensile capacity of the device through finite element simulation. However, given

this phenomenon that is not commonly seen, the author should describe the principle of this phenomenon in details.

Response 5: We thank Reviewer #1 for the opportunity to further describe two approaches of tuning the M_n sensors' characteristics *via* (1) stretching directions and (2) topographic design.

First, as discussed in **Response 4** earlier, FEA and *in situ* SEM studies were conducted to investigate the crack propagation behaviors of M_n sensors and their stain sensing performance under different stretching directions (including parallel, perpendicular, and 45°).

Second, as discussed in **Response 6** and **7** later, FEA and *in situ* SEM studies were conducted to investigate the crack propagation behaviors and the corresponding performance of M_n sensors with diverse topographic designs, including (1) selecting between heterogenous and homogeneous ps-MXene texturing, (2) varying the areal percentages of wrinkle-like region(s), and (3) altering the position and distribution of wrinkle-like region(s).

Revision made

Page 6–9, 27 of revised manuscript and Page S45–S51, S61–S64 of Supporting Information.

Comment 6: *The research on the wrinkles phenomenon in this paper focuses on G0 without wrinkles, GIF covering wrinkles, GIE of edge wrinkles, and G1M of the middle wrinkles, but the distribution of wrinkles is far more than these four kinds. Compared with this kind of research, the readers may be more interested in the proportion of the distribution area of wrinkles to the total area and the influence of the distribution position of wrinkles on the sensing performance of devices.*

Response 6: Per Reviewer #1's requests, several new types of M_n sensors were fabricated by (1) varying the areal percentages of wrinkle-like region(s) and (2) changing the distribution of wrinkle-like region(s).

Fig. S19 compares the strain sensing curves of various M_{w-p-w} sensors with different areal percentages of wrinkle-like regions. With higher areal percentages of wrinkle-like regions from 5% to 75%, the resulting M_{w-p-w} nanolayers exhibited more short/zigzag cracks under parallel stretching. The higher coverage of short/zigzag cracks prevented the conductive pathways from being completely cut off, leading to wider working windows of M_{w-p-w} sensors. Therefore, as the areal percentages increased from 5% to 75%, the linear working window of the resulting M_{w-p-w} sensor increased from 12–14% to 45–69%, respectively (**Fig. 3c**).

In addition, several new types of M_n sensors were fabricated by varying the positions and distribution of wrinkle-like region(s) (the areal percentage of wrinkle-like region(s) was controlled at 50%). To evaluate the effect of wrinkle texturing distribution on the strain sensing performance, two new kinds of M_n sensors, including M_{p-w} and $M_{w-p-w-p}$, were fabricated. The FEA results of M_{p-w} , M_{p-w-p} , M_{w-p-w} , and $M_{w-p-w-p}$ were shown in **Fig. S20**, showing that the wrinkle-like region(s) at the edge position(s) effectively reduced the overall localized strains of all four M_n sensors.

Fig. 3d compares the strain sensing curves of M_{p-w} and M_{p-w-p} , which have “one” wrinkle-like region. The average localized strains of M_{p-w} and M_{p-w-p} , sensors were quantified by *ImageJ* on the FEA results in **Figure S20**. The M_{p-w-p} nanolayer under 120% stretching showed an average localized strain of 32%, while the M_{p-w} nanolayer illustrated a lower average localized strain of 17%. With lower localized strains, the ϵ_{max} of a M_{p-w} sensor was characterized to be 37%, which was larger than the one of a M_{p-w-p} sensor ($\epsilon_{max} = 24\%$).

Fig. 3e compares the strain sensing curves of M_{w-p-w} and $M_{w-p-w-p}$, which have “two” wrinkle-like regions. The average localized strains of M_{p-w} and M_{p-w-p} , sensors were quantified by *ImageJ* on the FEA results in **Figure S20**. The M_{w-p-w} nanolayer under 120% stretching showed an average localized strain of 20% (quantified by *ImageJ* on the FEA result), while the $M_{w-p-w-p}$ nanolayer illustrated a higher average localized strain of 34%. With lower localized strains, the ϵ_{max} of a M_{w-p-w} sensor was characterized to be 50%, which was higher than the one of a $M_{w-p-w-p}$ sensor ($\epsilon_{max} = 32\%$). From both FEA results and strain sensing performance, setting the wrinkle-like region(s) at the edge position(s) was able to effectively reduce overall localized strains and increase M_n sensors’ ϵ_{max} .

Fig. S19 Strain sensing curves of M_{w-p-w} sensors with areal percentages of wrinkle-like region(s) from 5% to 75%.

Fig. 3 Tunable strain sensing characteristics of Mn sensors through topographic designs and stretching directions. (a) Relative resistance change (S_ε)–strain (ε) curves of M_n sensors under parallel stretching. Standard deviations were calculated based on three replicates. The composition of all M_n nanolayers was set at 85/10/5 (MXene/SWNT/PVA), and the thickness of all M_n nanolayers was controlled at 400 nm. (b) Relative resistance change (S_ε)–strain (ε) curves of M_n sensors under perpendicular stretching. (c) Linear working window(s) of the M_{w-p-w} sensors with different areal percentages of wrinkle-like region(s). (d) Strain sensing curves of M_{p-w-p} and M_{p-w} sensors. (e) Strain sensing curves of $M_{w-p-w-p}$ and M_{w-p-w} sensors. (f) Time-dependent relative resistance changes of a M_w sensor under various repeated uniaxial strains. (g) Reflection-contrast microscopy images for *in situ* observation of structural evolutions of a M_w nanolayer under repeated uniaxial strains. (h) Signal errors of M_n sensors. (i) Comparison of our M_n sensors with other strain sensors in the literature (in terms of gauge factor and linear working windows ($R^2 \geq 0.95$)). The gray color region represents the challenge region for current piezoresistive strain sensors to realize the gauge factors $>1,000$. The fabrication details of 12 M_n sensors are listed in Table S1.

Fig. S20 FEA simulation of four M_n nanolayers under 120% stretching.

Revision made

Page 9, 27 of revised manuscript and Page S23, S24 of Supporting Information.

Comment 7: *In Fig. 3a, it can be seen that the tensile property of GIF with wrinkles on the surface is greater than GIM with wrinkles in the middle but less than GIE with wrinkles on the edge. The authors should explain this phenomenon.*

Response 7: There are two categories of M_n nanolayers developed in this work: (1) the ps-MXene nanolayers with homogenous topographies (including M_p and M_w) and (2) the ps-MXene nanolayers with heterogeneous topographies (including M_{p-w-p} and M_{w-p-w}).

According to the FEA results in **Fig. S46**, the M_p and M_w nanolayers exhibited homogenous localized strain distribution profiles. Under parallel stretching, the M_p and M_w nanolayers exhibited similar in-plane crack densities and crack-to-width ratios (see definitions in **Methods**) between the edge and middle regions. On the other hand, from the FEA results in **Fig. S47**, the M_{p-w-p} and M_{w-p-w} nanolayers showed region-dependent and heterogeneous strain distribution profiles. Under parallel stretching, the M_{p-w-p} and M_{w-p-w} nanolayers showed distinct in-plane crack densities and crack-to-width ratios between the edge and middle regions. Because the FEA result could not reflect the regional mismatch of localized strains and the complexity of transition regions (between planar and wrinkle-like regions, **Fig. S8**), we only

compared the M_n sensors in the same category. The comparison (1) between M_p and M_w sensors (homogenous topographies) and (2) between M_{p-w-p} and M_{w-p-w} sensors (heterogenous topographies) are provided as follows.

First, the M_p and M_w sensors with homogenous topographies were compared. Based on the FEA results in **Fig. S46**, the M_p nanolayer under 120% stretching showed an average localized strain of 49% (quantified by *ImageJ*), while the M_w nanolayer demonstrated a much lower average localized strain of 20%. As a result, the M_p sensor demonstrated a much smaller ε_{max} of 6% than the M_w sensor ($\varepsilon_{max} = 39\%$).

Second, the M_{p-w-p} and M_{w-p-w} sensors with heterogenous topographies were compared. Based on the FEA results in **Fig. S47**, the M_{p-w-p} nanolayer under 120% stretching showed an average localized strain of 32% (quantified by *ImageJ*), while the M_{w-p-w} nanolayer demonstrated a lower average localized strain of 20%. As a result, the M_{w-p-w} sensor demonstrated a much smaller ε_{max} of 24% than the M_{p-w-p} sensor ($\varepsilon_{max} = 50\%$).

Fig. S8 High-resolution SEM images of the transition zone of a M_{p-w-p} nanolayer, where the wrinkle wavelength gradually increased.

Fig. S46 Performance comparison between M_p and M_w sensors. (a) FEA simulation and *in situ* SEM image of a M_p nanolayer under parallel stretching. **(b)** FEA simulation and *in situ* SEM image of a M_w nanolayer under parallel stretching. **(c)** Strain sensing curves of M_p and M_w sensors.

Fig. S47 Performance comparison between M_{p-w-p} and M_{w-p-w} sensors. (a) FEA simulation and *in situ* SEM image of a M_{p-w-p} nanolayer under parallel stretching. (b) FEA simulation and *in situ* SEM image of a M_{w-p-w} nanolayer under parallel stretching. (c) Strain sensing curves of M_{p-w-p} and M_{w-p-w} sensors.

Revision made

Page 8 of revised manuscript and Page S50, S51, S63, S64 of Supporting Information.

Comment 8: *As for the naming method in this article, write "the planar ps-MXene nanolayers (named as G0)" in line 1 on page 6, but the "G" after the abbreviation is not related to the original word, which may cause inconvenience to readers. Authors should name abbreviations*

that are easier to understand.

Response 8: We thank Reviewer #1's suggestion, and we have changed our naming method. For instance, G_{I-E} is now named as M_{w-p-w} , where M indicates ps-MXene nanolayer, the subscripts $w-p-w$ indicates the topographic design, w refers to the wrinkle-like region, p refers to the planar region. G_{I-M} was thus named M_{p-w-p} , G_0 became M_p , and G_{I-F} became M_w .

We have updated all of the sensors' names in the revised manuscript and Supporting Information.

Comment 9: *In Figure 3d, the "signal errors" value of G_0 is much larger than the other three sensors, which is interpreted by the author as "where large surface cracks were randomly generated under strains." I think this may be caused by the poor dispersion of nanomaterials. According to the author's explanation, the "signal errors" value of G_{IF} should be much higher than that of the other three materials because of its most folded distribution.*

Response 9: We thank Reviewer #1's careful review. We would like to address that high signal error of M_p sensor was not due to the poor dispersion of nanomaterials. As we used the same dispersion for all M_n sensors, their signal errors still varied. According to **Eqn. 1** and **4**, the signal error was mainly determined by the performance repeatability of multiple M_n sensor replicates from the same fabrication recipe,

$$S_\varepsilon = \frac{R_\varepsilon - R_0}{R_0} \quad (1)$$

$$\text{Signal Error} = \frac{\sigma_{S_\varepsilon}}{\varepsilon} \quad (4)$$

, where S_ε is the relative resistance difference, σ_{S_ε} is the standard deviation of S_ε from M_n sensor replicates under an applied strain (ε). The large signal error resulted from large S_ε variations from multiple M_n sensor replicates.

According to **Movie S1**, the M_p nanolayer exhibited uncontrolled crack propagation behaviors, showing that the large cracks were randomly generated under strains. In comparison, according to **Movie S2** and **S3**, the M_w nanolayer demonstrated a more controllable crack propagation fashion, showing that the zigzag cracks were constrained along the valleys of periodic wrinkles and emerged repeatedly under strains. Therefore, with one or more wrinkle-textured region(s), the M_w , M_{w-p-w} , and M_{p-w-p} sensors (three replicates) demonstrate more consistent strain sensing performance and thus lower signal errors <10% (as shown in **Fig. 3h**). On the other hand, with only planar nanolayers, the M_p sensors (three replicates) exhibited the largest signal errors >50%.

We have added more discussion in our revised manuscript to make this part clearer.

Revision made

Page 10, 11 of revised manuscript.

Comment 10: In Figure S17, the GIF sensor has worked continuously for 2000 times under 30% strain, but this demonstrated lifetime is far from enough. Human joints typically moved thousands of times a day. I think the author should cycle test all types of sensors used for integration to determine their application potential.

Response 10: Per Reviewer #1's request, we have tested the cycling performance of all types of M_n sensors used in this work (including M_p , M_{p-w-p} , M_w , M_{w-p-w}) up to 20,000 cycles. As shown in Fig. S21 to S24, the relative resistance changes of all M_n sensors remained stable during the tests.

Fig. S21 Cycling test of a M_p sensor under 5% strain for 20,000 cycles.

Fig. S22 Cycling test of a M_{p-w-p} sensor under 15% strain for 20,000 cycles.

Fig. S23 Cycling test of a M_w sensor under 25% strain for 20,000 cycles.

Fig. S24 Cycling test of a M_{w-p-w} sensor under 40% strain for 20,000 cycles.

Revision made

Page 10 of revised manuscript S25–S28 of Supporting Information.

Comment 11: *Was the integration process for the sensing devices used in this work an innovative integration method? Or just a universal method?*

Response 11: In this work, we achieved several milestones, including (1) topographic design of M_n sensors, (2) integration of edge computing chip, and (3) in-sensor machine learning for full-body avatar reconstruction. **Table 1** compares our edge sensor module with the state-of-the-art works in the literature in terms of tactile sensor features and application scenes. Our edge sensor module showed advances in terms of tunable sensor design and versatile

integration method. As mentioned by multiple important review articles (*Adv. Mater.* 2022, 2107902; *Nat. Electron.* 2022, 5, 142–156; *Nat. Electron.* 2020, 3, 664–671; *Adv. Funct. Mater.* 2021, 31, 2008807), an urgent challenge in wearable sensors is to enable efficient transmission of large amounts of sensor data followed by in-sensor machine learning with power-efficient local computing. Multidisciplinary efforts are required to address this challenge by advancing hardware/software development and optimizing the sensor/circuit interfaces. In this work, our edge sensor module was carefully designed to achieve full-body motion monitoring coupled with edge data processing and in-sensor machine learning, which has not been reported before. We believe that the integration method (i.e., wearable/stretchable sensors with edge computing chip(s)) demonstrated in this work is highly versatile and can benefit the advances in other fields, including wearable performance devices in sports and underwater soft robots.”

Revision made

Page 16 of revised manuscript.

Table 1. Comparison of this work with other wearable sensor systems for human motion monitoring.

Reference	Sensor Type	No. of Sensor Type (Location)	No. of Sensors	Wireless Data Transmission	Signal Acquisition	Motion Classification	Avatar Reconstruction	Edge Computing
Jeong, H. et al. Proc. Natl. Acad. Sci. U. S. A. 2021. ¹	Inertial sensor	1 (Arms, legs, head, chest)	10	Yes	Single-Channeled	×	√	×
Tautges, J. et al. ACM Trans. Graph. 2011. ¹⁹	Inertial sensor	1 (Arms and legs)	6	Yes	Single-Channeled	×	√	×
Luo, Y. et al. Nat. Electron. 2021. ⁸	Strain sensor	1 (All joints)	>200	No	Multi-Channeled	√	×	×
Wang, M. et al. Nat. Electron. 2020. ²⁶	Strain sensor	1 (Hands)	5	No	Multi-Channeled	√	×	×
Zhou, Z. et al. Nat. Electron. 2020. ²⁷	Strain sensor	1 (Hands)	5	Yes	Multi-Channeled	√	×	×

Lin, M. et al. Adv. Mater. 2021. ⁶⁷	Strain sensor	1 (Wrist and knees)	4	No	Single-Channeled	×	×	×
Kim, J. H. et al. Adv. Funct. Mater. 2021. ²	Strain sensor	1 (Elbows and knees)	4	No	Single-Channeled	×	×	×
Yang, Z. et al. ACS Nano 2018. ⁴⁰	Strain sensor	1 (All joints)	8	No	Single-Channeled	×	×	×
Sun, H. et al. ACS Appl. Mater. Interfaces 2019. ⁶⁸	Strain sensor	1 (Knees)	2	No	Single-Channeled	×	×	×
Ma, J. et al. ACS Appl. Mater. Interfaces 2019. ⁶⁹	Strain sensor	1 (Elbows and knees)	4	No	Single-Channeled	×	×	×
Yang, H. et al. ACS Nano 2020. ⁷	Strain sensor	1 (Hands)	2	Yes	Single-Channeled	×	×	×
This Work	Piezoresistive Sensor	4	7	Yes	Multi-Channeled	√	√	√

(Back waist,
shoulders elbows,
knees)

Reviewer #2

Comment 1: *In the manuscript, wearable $Ti_3C_2T_x$ MXene sensor modules are fabricated with in-sensor machine learning (ML) models, either functioning via wireless streaming or edge computing, for full-body motion classifications and avatar reconstruction. Through topographic design on piezoresistive nanolayers, the wearable strain sensor modules exhibited ultrahigh sensitivities within the working windows and meet all joint deformation ranges. By integrating the wearable sensors with a ML chip, an edge sensor module is fabricated, enabling in-sensor reconstruction of high-precision avatar animations that mimic continuous full-body motions with 92.6% accuracy. However, the following problems and issues should be addressed.*

Response 1: We thank Reviewer #2 for his/her careful review and valuable comments. We have made point-by-point responses below.

Comment 2: *In the manuscript, a lot of mechanical and electrical characterizations of the wrinkle-like MXene textures should be provided.*

Response 2: We appreciate Reviewer #2's suggestion. In the revised manuscript, we supplemented the mechanical measurements of all M_n sensors and the electrical characterizations of wrinkle-like ps-MXene textures, including (1) the stress–strain curves of all M_n sensors (see **Fig. S26**), (2) the electrical conductivity of a planar ps-MXene nanolayer (confirmed to be $2,479 \text{ S cm}^{-1}$ at the MXene/SWNT/PVA ratio of 85/10/5), and (3) the electrical resistances of all kinds of wrinkle-like ps-MXene textures (see **Fig. S9**).

Fig. S26 Stress–strain curves of all M_n sensors. As the nanolayer thickness was controlled at 400 nm, all M_p , M_{p-w-p} , M_w , and M_{w-p-w} sensors showed similar Young's moduli of ca. 150 kPa, which were higher than a bare VHB™ tape (106 kPa).

Fig. S9 Electrical resistances of all kinds of M_n nanolayers. The lower resistances of M_{p-w-p} , M_w , and M_{w-p-w} nanolayers were attributed to the peak contacts between dense wrinkles, which shortened the electrical pathways. The width of all M_n nanolayers was controlled to be 1 cm.

Revision made

Page 6, 10 of revised manuscript and Page S13, S30 of Supporting Information.

Comment 3: *In the manuscript, the cyclic stability of the sensor should be considered.*

Response 3: Per Reviewer #2's request, we have tested the cycling performance of all M_n sensor types demonstrated in this work (including M_p , M_{p-w-p} , M_w , M_{w-p-w}) up to 20,000 cycles. As shown in **Fig. S21–S24**, the relative resistance changes of all M_n sensors remained stable during the tests.

Revision made

Page 10 of revised manuscript S25–S28 of Supporting Information.

Comment 4: *In the application part of the sensors, the significance of mimicking continuous full-body motions should be provided?*

Response 4: We thank Reviewer #2's insightful suggestion. In the revised manuscript, we have added more discussion regarding (1) the importance of monitoring continuous full-body motions and (2) the advantages of integrating wearable M_n sensors with edge computing chip(s).

“Precise monitoring of continuous human motions has long been a primary center for various applications, such as gesture/gait recognition, animation production, remote healthcare, and virtual reality. During the coronavirus pandemic, the technologies for sensing delicate body motions (e.g., trembling, shivering) become necessary, as the doctors can monitor the patients' symptoms in real time. Also, the human motion sensing technologies have significant impacts in various industries, including sports, healthcare, and gaming entertainment.

In this work, we achieved several milestones, including (1) topographic design of M_n sensors, (2) integration of edge computing chip, and (3) in-sensor machine learning for full-body avatar reconstruction. **Table 1** compares our edge sensor module with the state-of-the-art works in the literature in terms of tactile sensor features and application scenes. Our edge sensor module showed advances in terms of tunable sensor design and versatile integration method. As mentioned by multiple important review articles (*Adv. Mater.* 2022, 2107902; *Nat. Electron.* 2022, 5, 142–156; *Nat. Electron.* 2020, 3, 664–671; *Adv. Funct. Mater.* 2021, 31, 2008807), an urgent challenge in wearable sensors is to enable efficient transmission of large amounts of sensor data followed by in-sensor machine learning with power-efficient local computing. Multidisciplinary efforts are required to address this challenge by advancing hardware/software development and optimizing the sensor/circuit interfaces. In this work, our edge sensor module was carefully designed to achieve full-body motion monitoring coupled with edge data processing and in-sensor machine learning, which has not been reported before. We believe that the integration method (i.e., wearable/stretchable sensors with edge computing chip(s)) demonstrated in this work is highly versatile and can benefit the advances in other fields, including wearable performance devices in sports and underwater soft robots.”

Revision made

Page 13, 14, 16 of revised manuscript.

Comment 5: *In Fig.3a, why the relative resistance is changed by the thickness of G_n nanolayers?*

Response 5: In the revised manuscript, additional *in situ* electron microscopic studies have been conducted to investigate the effect of nanolayer thicknesses on the M_n sensors’ crack propagation behaviors and strain sensing performance. Taking the M_w sensor as an example in **Fig. S48**, by increasing the nanolayer thicknesses from 400 to 800 nm, the average wrinkle wavelength of M_w nanolayer increased from ca. 7 to 13 μm . With different wrinkle wavelengths, the M_w sensors exhibited different crack propagation behaviors. **Fig. S49a** recorded the *in situ* SEM images of two M_w sensors (with 800- and 400-nm-thick nanolayers) under parallel stretching. As summarized in **Fig. S49b**, the crack-to-width ratio (see definition in **Methods**) of the M_w sensor with a thicker nanolayer increased faster than the one with a thinner nanolayer. In addition, in **Fig. S49c**, the maximal crack density (see definition in **Methods**) of the M_w sensor with a thinner nanolayer was 1,920 $\mu\text{m mm}^{-2}$, larger than the one with a thicker nanolayer (1,260 $\mu\text{m mm}^{-2}$).

Fig. S48 By increasing the nanolayer thickness from 400 to 800 nm, the wrinkle wavelength of M_w nanolayer increased from ca. 7 to 13 μm . The M_w nanolayers kept the same MXene/SWNT/PVA ratio of 65/30/5.

Fig. S49 Crack propagation of M_w nanolayers with varying thicknesses under various strains. (a) SEM images of M_w nanolayers with varying thicknesses under various strains. (b) Crack-to-width ratios and (c) crack densities of M_w nanolayers with different thicknesses. The M_w nanolayers kept the same MXene/SWNT/PVA ratio of 65/30/5.

Revision made

Page 11 of revised manuscript S52, S53, S64 of Supporting Information.

Comment 6: *In the manuscript, the innovation and mimic continuous full-body motions should be provided clearly.*

Response 6: As discussed earlier in **Response 4**, we have added more discussion regarding (1) the importance of monitoring continuous full-body motions and (2) the advantages of integrating wearable M_n sensors with edge computing chip(s).

“Precise monitoring of continuous human motions has long been a primary center for various applications, such as gesture/gait recognition, animation production, remote healthcare, and virtual reality. During the coronavirus pandemic, the technologies for sensing delicate body motions (e.g., trembling, shivering) become necessary, as the doctors can monitor the patients’ symptoms in real time. Also, the human motion sensing technologies have significant impacts in various industries, including sports, healthcare, and gaming entertainment.

In this work, we achieved several milestones, including (1) topographic design of M_n sensors, (2) integration of edge computing chip, and (3) in-sensor machine learning for full-body avatar reconstruction. **Table 1** compares our edge sensor module with the state-of-the-art works in the literature in terms of tactile sensor features and application scenes. Our edge sensor module showed advances in terms of tunable sensor design and versatile integration method. As mentioned by multiple important review articles (*Adv. Mater.* 2022, 2107902; *Nat. Electron.* 2022, 5, 142–156; *Nat. Electron.* 2020, 3, 664–671; *Adv. Funct. Mater.* 2021, 31, 2008807), an urgent challenge in wearable sensors is to enable efficient transmission of large amounts of sensor data followed by in-sensor machine learning with power-efficient local computing. Multidisciplinary efforts are required to address this challenge by advancing hardware/software development and optimizing the sensor/circuit interfaces. In this work, our edge sensor module was carefully designed to achieve full-body motion monitoring coupled with edge data processing and in-sensor machine learning, which has not been reported before. We believe that the integration method (i.e., wearable/stretchable sensors with edge computing chip(s)) demonstrated in this work is highly versatile and can benefit the advances in other fields, including wearable performance devices in sports and underwater soft robots.”

Revision made

Page 13, 14, 16 of revised manuscript.

Reviewer #3

Comment 1: *In this manuscript, the G_n sensors were fabricated with Ti3C2Tx MXene, SWNTs and PVA. Through topographic design during localized thermal contraction, the G_n sensor modules exhibited ultrahigh sensitivities within the working windows that met various joint deformation ranges. Neural networks were used to fuse the sensor data for full-body motion classifications and avatar reconstruction. Edge sensor module with ML chip was used to reduce wireless data flow and power consumption. The work is interesting, but the following questions need to be concerned and addressed before considering the publication:*

Response 1: We thank Reviewer #3 for his/her careful review and valuable comments. We have made point-by-point responses for each comment below.

Comment 2: *The strain sensor needs to be further tested to evaluate its performance, such as hysteresis, response time.*

Response 2: Per Reviewer #3's request, we have supplemented additional experiments to characterize the hysteresis curves of all M_n sensors (**Fig. S27**) and the response time of a M_p sensor (**Fig. S25**).

The hysteresis of a M_n sensor ($U_{hysteresis}$) was quantified by measuring the maximal signal difference between the stretching and releasing processes, as defined in **Eqn. 3**,

$$U_{hysteresis} = \text{Max}|S_{stretching} - S_{releasing}| \quad (3)$$

, where $S_{stretching}$ is the relative resistance change signal, $(R-R_0)/R_0$, of a M_n sensor during the stretching process, and $S_{releasing}$ is the relative resistance change of a M_n sensor during the relaxation process. Based on **Fig. S27**, the hysteresis of M_p , M_{p-w-p} , M_w , and M_{w-p-w} sensors were calculated as 18, 27, 28, and 31, respectively, which corresponded to their testing strain of 5%, 15%, 25%, and 40%. This trend was reasonable, as the hysteresis of a VHB tape increased with the applied strain (*Macromol. Mater. Eng.* 2015, 300, 99-106; *Polym. Test.* 2022, 109, 107557.).

As illustrated in **Fig. S25**, the response times of a M_p sensor in its stretching and relaxation processes were characterized to be 0.064 and 0.082 seconds, respectively.

Fig. S27 Hysteresis curves of M_n sensors.

Fig. S25 Response times of a M_p sensor in the stretching and relaxation processes.

Revision made

Comment 3: Please provide enlarged profiles of step responses under dynamic stretching and releasing ($> 30\%$ strain). Peak shoulders appear when the strain $> 25\%$ as shown in Fig. 3b. Please clarify.

Response 3: Per Reviewer #3's request, as shown in Fig. S57, we provided the enlarged profiles of step responses of a M_w sensor during the cycling test (under 32% repeated strains). The flattened plateaus on the signal peaks resulted from the default mode of our tensile tester, as the movement of tensile grips slowed down to transit from stretching (open grips) to relaxation (close grips).

Fig. S57 Time-dependent relative resistance changes of a M_w sensor during a cycling test (under 32% repeated strains).

Revision made

Page 9, 10 of revised manuscript.

Comment 4: The reviewer suggests to replace “prediction” by “estimation” or “determination” in the manuscript, as “prediction” generally refers to predictive estimation ahead of time. If so, the leading time should be given?

Response 4: We thank Reviewer #3 for his/her valuable suggestion. We have replaced “prediction” by “determination” in the revised manuscript.

Comment 5: The authors mentioned “By continuously receiving the voltage data, the ML chip with in-sensor CNN model enabled real-time and high-accuracy prediction of 15 avatar joint locations.” How to determine the fixed time length for the grid-like input data of CNN? The

time delay would be serious if the time length was long.

Response 5: We thank Reviewer #3 for his/her insightful comment. We want to clarify that, in this work, the CNN kernel size was selected using Bayesian optimization (see reference in **GitHub**: <https://github.com/fmfn/BayesianOptimization>). Bayesian optimization enables hyperparameter tuning without the need for grid search. Additionally, for the prediction of full-body motion movements, the Bayesian optimizer tunes the time length of newly input sensor data by considering the history of previous sensor data. As shown in **Fig. S54**, the First-in-First-out (FIFO) register stores the previously input sensor data and bundle them with the new input data, and the data points (past + current) are used to predict the avatar movement. In this work, the end-to-end regression with CNN took less than 1 millisecond.

Fig.54 Implementation of FIFO register for real-time computation using a CNN model.

Revision made

Page S58, S68, S69 of Supporting Information.

Comment 6: *The proposed method for evaluating the estimation accuracy of joint locations (i.e. eq. 8) is incorrect and unreasonable. The authors proposed a relative index by dividing location errors by absolute location T_i . The absolute location T_i is self-defined coordinate position, cannot be as the denominator. The accuracy would be tuned to be very high if setting a large value of T_i . The accuracy should be evaluated using absolute location error with physical unit (e.g. mm or cm) rather than relative expression.*

Response 6: We appreciate Reviewer #3's suggestion. Indeed, using relative location errors may be misleading for the readers, so we have re-calculated the absolute location errors in the unit of "cm" in the revised manuscript.

"The determination error of in-sensor avatar reconstruction was calculated by measuring the difference between the real joint locations (extracted from camera-recorded videos, **Table S5** in **GitHub**) and the determined joint locations (computed by edge sensor module, **Table S6** in **GitHub**). The average determination error is defined in **Eqn. 8**,

$$\text{Average Determination Error} = \frac{1}{15} \times \frac{170 \text{ cm}}{830} \times \sum_{i=0}^{14} \sum_0^t |P_t^i - T_t^i| \quad (8)$$

, where P_t^i is the CNN-determined location of i^{th} joint at the time t , T_t^i is the camera-recorded locations of i^{th} joint at the time t , 170 cm is the volunteer’s physical height, and 830 is the corresponding avatar height in the virtual coordinate system, and 15 is the number of monitored joints. By comparing 15 joint locations (**Fig. 5d** and **S35**), the average avatar determination error was calculated to be 3.5 cm.”

Revision made

Page 14 of revised manuscript.

Comment 7: *The present work only concerned two-dimensional reconstruction using camera recorded video as reference data. However, human body/limb motions are actually three-dimensional movements. How to solve this problem?*

Response 7: We thank Reviewer #3’s valuable comment. As shown in **Fig. S56** and **Movie S5**, by adopting the “MotioNet” method from the computer vision field (see more details in Reference: *ACM Transactions on Graphics*, 2020, 40, 1–15.), a 3D avatar was constructed based on the 2D avatar results determined by the edge sensor module. The working mechanism of MotioNet is similar to the OPEN POSE program. First, MotioNet extracted the full-body motions from the recorded videos and estimated 15 joint locations of a volunteer in x , y , and z coordinates. As shown in **Fig. S56**, a stationary 3D avatar with 15 joints was then constructed. Afterward, based on the M_n sensor database, the CNN model was re-trained to output the joint locations in x , y , and z coordinates, which were utilized by MotioNet to re-construct the 3D avatar’s animation.

Fig. S56 Reconstruction of a 3D avatar via MotioNet method.

Revision made

Page 15 of revised manuscript and Page S60, S70 of Supporting Information.

Comment 8: *In Figure 2c, it seems that the marking of texture region and planar region of G1-E is not correct? Please check.*

Response 8: We appreciate Reviewer #3's careful review. We have corrected **Fig. 2d** in the revised manuscript.

Revision made

Page 26 of revised manuscript.

Comment 9: *What are the resistance value and electric conductivity of G_n sensors under non-tensile strain?*

Response 9: In the revised manuscript, we supplemented the electrical characterizations of all M_n sensors and the wrinkle-like ps-MXene textures, (1) the electrical conductivity of a planar ps-MXene nanolayer (confirmed to be $2,479 \text{ S cm}^{-1}$ at the MXene/SWNT/PVA ratio of 85/10/5), and (2) the electrical resistances of all kinds of wrinkle-like ps-MXene textures (see **Fig. S9**).

Fig. S9 Electrical resistances of all kinds of M_n nanolayers. The lower resistances of M_{p-w-p} , M_w , and M_{w-p-w} nanolayers were attributed to the peak contacts between dense wrinkles, which shortened the electrical pathways. The width of all M_n nanolayers was controlled to be 1 cm.

Revision made

Page 6 of revised manuscript and Page S13 of Supporting Information.

Comment 10: *In Page 10, the authors mentioned “Our wireless sensor module demonstrated a high data transmission rate (S) at ca. 450 bps”. This transmission rate is very slow and cannot be expressed as “high”. Usually, the transmission rate of Bluetooth can reach about 1 Mbps.*

Response 10: We thank Reviewer #3's careful review. We have deleted the term “high” in the revised manuscript.

“Our wireless sensor module demonstrated a data transmission rate (S) at ca. 450 bps (bits per second).”

Revision made

Page 12 of revised manuscript.

Comment 11: *In Page 11, the authors also mentioned “Our wireless sensor module maintained its high S values above 400 bps within 80 meters (Fig. S24a), across 4 cement brick walls (one wall thickness ~20 cm) (Fig. S24b), and after 100-hour continuous operation (Fig. S25).” This performance is attributed to the commercial Bluetooth HC-06, not the contribution of the proposed sensor module. The reviewer suggests to modify the description and delete Fig. S24.*

Response 11: We thank Reviewer #3’s suggestion. We have modified the description and deleted **Fig. S24** and **S25** in the revised manuscript.

“As we adopted a commercial Bluetooth HC-06, our wireless sensor module maintained its S value above 400 bps in a broad space (>80 meters) and long-term (>100 hours) continuous operation.”

Revision made

Page 13 of revised manuscript.

Comment 12: *There are problematic issues in the statement of “Each G_n sensor was connected in series with standard resistors, and the resistor value was specifically selected to be 100 k Ω to ensure wireless data transmission with high accuracy.”, Fig. 4g, Fig. S22 and Methods “Circuit optimization of a wireless sensor module.”*

The mentioned “transmission error” is not caused by the standard resistor value, it is due to the inherent input resistor of ADC and can be eliminated by adding the voltage follower.

Response 12: We thank Reviewer #3 for noting this issue in the manuscript. As shown in **Fig. S50**, the voltage across a M_n sensor was measured by an ADC unit. As the applied strains increased, the resistance of a M_n sensor increased accordingly from 0.1 to 100 k Ω , and the input impedance of an ADC unit was about 1 M Ω (see ADC data sheet in **GitHub**: <https://github.com/Haitao008/Supporting-Tables>). As a result, when the resistances of M_n sensors increased under strains, the total resistance that consisted of a parallel impedance combination of an ADC unit and M_n sensors changed from 0.09 to 90.9 k Ω , leading to the resistance deviations of M_n sensors from 0.1 to 9.0%. It is worth noting that, as shown in **Eqns. S1, 12** and the code screenshot in **Fig. S58**, the input impedance of the ADC unit had been considered and included when calculating the output resistance values of a M_n sensor from the ADC-recorded voltage values as well as estimating the transmission errors. In other words, when calculating the transmission error, we had already considered the resistance deviations induced by the ADC unit. We have revised **Eqn. 5** in the revised manuscript.

$$R_{output} = \frac{V_{ADC} \times R_{standard} \times R_{ADC}}{5 \times R_{ADC} - V_{ADC} \times (R_{standard} + R_{ADC})} \quad (\text{S1})$$

, where R_{output} is the estimated resistance of a M_n sensor, V_{ADC} is ADC-measured voltage across a M_n sensor, 5 is the applied voltage, R_{ADC} is the input impedance of an ADC (i.e., 1 M Ω), and $R_{standard}$ is the resistance of an integrated standard resistor.

Under an applied voltage of 5.0 V, the voltage read from ADC across a M_n sensor is dependent on the value of the standard resistor connected in series (see the circuit in **Fig. 4g**). **Table S7** compares the voltages read from ADC across a M_n sensor (with strain-dependent resistances), which was connected in series with a 5- or 100-k Ω standard resistor. As shown in **Table S7**, if a 100-k Ω standard resistor was connected, the voltages read from the ADC across a M_n sensor increased from 1.61 to 2.26 V, when the M_n sensor's resistance increased with strains from 50 to 90 k Ω . On the other hand, if a 5-k Ω standard resistor was connected, the voltage read from ADC across a M_n sensor only increased from 4.52 to 4.71 V, when the M_n sensor's resistance increased with strains from 50 to 90 k Ω . The small voltage changes were affected severely by the noise signals and caused the fluctuations of ADC-measured voltages, leading to the wrong estimations of M_n sensor resistance values. The transmission error was mainly because of the noise associated with reading voltage with discrete ADC and wireless modules.

$$\text{Wireless Transmission Error} = \frac{|R_{input} - R_{output}|}{R_{input}} \quad (12)$$

, where R_{input} is the resistance value of a M_n sensor measured directly by a digital multimeter, and R_{output} is the output resistance value of a M_n sensor wirelessly transmitted and converted from the ADC-recorded voltage value.

It is worthy to mention that, by integrating tunable impedance into the circuit, we could avoid the selection of optimal standard resistors to accompany M_n sensors and further achieve tight circuit integration (with M_n sensors, ADC, and wireless/edge computing modules), which can reduce the noise associated with discrete devices.

Fig. S50 Equivalent circuit of the connections among a M_n sensor, ADC, and standard resistor.

```

function [R_output] = cvtresist(voltage)
    R_standard = 100000;
    R_ADC = 1000000;
    R_output = (voltage*R_standard*R_ADC)/(5*R_ADC-voltage*(R_standard+R_ADC));
end

```

Fig. S58 Code screenshot of the estimation of a M_n sensor resistance converted from the ADC-recorded voltage.

Table S7 Comparison of ADC-read voltages using different standard resistor values of 5 k Ω and 100 k Ω .

Resistance of a M_n Sensor	Voltage read from ADC (calculated with 5-k Ω standard resistance)	Voltage read from ADC (calculated with 100-k Ω standard resistance)
50 k Ω	4.52 V	1.61 V
60 k Ω	4.60 V	1.81 V
70 k Ω	4.65 V	1.98 V
80 k Ω	4.68 V	2.13 V
90 k Ω	4.71 V	2.26 V

Revision made

Page 12 of revised manuscript and Page S54, S65, S66, S72 of Supporting Information.

Comment 13: *Some definitions are unclear. For example, the “Feed-in resistance” in Fig S22 requires the reader to guess what resistance it refers to, in the statement “where R_{input} is the resistance of the standard resistors (from 0.1 to 100 k Ω) used to replace G_n sensors in the wireless sensor module. The wireless sensor module with 100-k Ω standard resistors showed low transmission errors of 3%,” the two “standard resistors” do not actually refer to the same resistance, which may confuse readers.*

It is suggested that the authors should improve the circuit design and clarify the confusion of the circuit description.

Response 13: We thank Reviewer #3’s valuable suggestion. We agree with Reviewer #3 that using standard resistors to determine “wireless transmission error” is confusing to the readers. Therefore, we have re-measured the wireless transmission error directly using the M_n sensors under different strains.

Similar results were obtained in the revised manuscript. As shown in **Fig. S32**, with 100-k Ω standard resistors, the wireless sensor module exhibited low and strain-stable transmission errors <4%. In the other words, a M_{w-p-w} sensor under 48% strain (with a 39.1-k Ω resistance) was read as 37.7 k Ω after wireless data transmission, showing a transmission error of 3.6%. On the other hand, with 5-k Ω standard resistors, the wireless sensor module exhibited

higher and strain-dependent transmission errors (the average error >20%). For instance, a M_{w-p-w} sensor under 48% strain (with a 39.1-k Ω resistance) was read as 13.3 k Ω after wireless data transmission, showing a transmission error of 65%.

In the revised manuscript, we have modified the circuit description, **Fig. S32**, and figure caption to make this part clearer.

“Second, standard resistors were used as the regulator to enable high measurement accuracy with low wireless transmission errors, which were calculated by following **Eqn. 12**,

$$\text{Wireless Transmission Error} = \frac{|R_{input} - R_{output}|}{R_{input}} \quad (12)$$

, where R_{input} is the resistance value of a M_n sensor measured directly by a digital multimeter, and R_{output} is the output resistance value of a M_n sensor wirelessly transmitted and converted from the ADC-recorded voltage value.

In this work, we measured wireless transmission error by using the M_n sensors under different strains. As shown in **Fig. S32**, with 100-k Ω standard resistors, the wireless sensor module exhibited low and strain-stable transmission errors <4%. In the other words, a M_{w-p-w} sensor under 48% strain (with a 39.1-k Ω resistance) was read as 37.7 k Ω after wireless data transmission, showing a transmission error of 3.6%. On the other hand, with 5-k Ω standard resistors, the wireless sensor module exhibited higher and strain-dependent transmission errors (the average error >20%). For instance, a M_{w-p-w} sensor under 48% strain (with a 39.1-k Ω resistance) was read as 13.3 k Ω after wireless data transmission, showing a transmission error of 65%.”

Fig. S32 Wireless transmission errors of wireless sensor modules. (a) Circuit of a wireless sensor module. (b) Low transmission errors (average errors <4%) were observed in the wireless sensor module with 100-k Ω standard resistors. (c) High transmission errors (average errors >20%) were observed in the wireless sensor module with 5-k Ω standard resistors. Here, the wireless transmission error was measured by using a M_{w-p-w} sensor under different strains.

Revision made

Page 21 of revised manuscript and Page S36 of Supporting Information.

Comment 14: *The authors mentioned: “The freestanding nanolayers in ethanol were carefully transferred onto VHB tapes followed by overnight drying.”*

According to our knowledge, the rebound performance of VHB tape is very poor and the hysteresis is very large. How to ensure the measurement accuracy of the sensor during movement using VHB tape? The reviewer suggests to provide the strain response profiles of the sensor under dynamic stretching and releasing for several cycles? Hysteresis of the sensor needs to be tested and evaluated.

Response 14: We agree with Reviewer #3 that VHBTM tape demonstrated superior mechanical stability yet was also accompanied by large hysteresis. In the revised manuscript, the hysteresis curves of all M_p sensors (including M_p , M_{p-w-p} , M_w , and M_{w-p-w} sensors) are shown in **Fig. S27**.

The hysteresis of a M_n sensor ($U_{hysteresis}$) was quantified by measuring the maximal signal difference between the stretching and releasing processes, as defined in **Eqn. 3**,

$$U_{hysteresis} = \text{Max}|S_{stretching} - S_{releasing}| \quad (3)$$

, where $S_{stretching}$ is the relative resistance change signal, $(R-R_0)/R_0$, of a M_n sensor during the stretching process, and $S_{releasing}$ is the relative resistance change of a M_n sensor during the relaxation process. Based on **Fig. S27**, the hysteresis of M_p , M_{p-w-p} , M_w , and M_{w-p-w} sensors were calculated as 18, 27, 28, and 31, respectively, which corresponded to their testing strain of 5%, 15%, 25%, and 40%. This trend was reasonable, as the hysteresis of a VHB tape increased with the applied strain (*Macromol. Mater. Eng.* 2015, 300, 99-106; *Polym. Test.* 2022, 109, 107557.).

In addition, in **Fig. S21** to **S24**, we tested the cycling performance of all M_n sensors for 20,000 cycles, and all M_n sensors demonstrated stable hysteresis of their resistance changes during the repeated stretching and relaxation cycles. Furthermore, the CNN model was trained by using the entire resistance curves of M_n sensors with hysteresis as the training data. Therefore, the trained CNN model was able to calibrate the hysteresis of all M_p sensors and ensured a high accuracy of motion reconstruction (with the average avatar determination error of 3.5 cm).

Fig. S27 Hysteresis curves of M_p , M_{p-w-p} , M_w , and M_{w-p-w} sensors under uniaxial strains.

Fig. S21 Cycling test of a M_p sensor under 5% strain for 20,000 cycles.

Fig. S22 Cycling test of a M_{p-w-p} sensor under 15% strain for 20,000 cycles.

Fig. S23 Cycling test of a M_w sensor under 25% strain for 20,000 cycles.

Fig. S24 Cycling test of a M_{w-p-w} sensor under 40% strain for 20,000 cycles.

Revision made

Page 10 of revised manuscript and Page S25–S28, S31 of Supporting Information.

Comment 15: *The proposed devices still have the limitations of inconvenient/uncomfortable wearing, wires spreading all over human body and limbs, which seriously affects the user's movement experience.*

Response 15: We appreciate Reviewer #3's valuable comment. To increase the comfort levels of wearing the M_n sensor modules, two approaches have been developed in the revised manuscript. First, as shown in **Fig. S55a**, the copper wires were sewed into the fabrics to avoid circuit disorders over the human body and limbs. Second, the M_n sensors were further stabilized on the clothing using commercial hook-and-loop fasteners, which provided sufficient mechanical stability and became convenient to be attached/detached or adjusted (**Fig. S55b**).

Fig. S55 Increasing comfort level of wearing sensor modules. (a) Copper wires were sewed into the fabrics to avoid circuit disorders. (b) Commercial hook-and-loop fasteners were used to stabilize the M_n sensors on the clothing.

Revision made

Page 15 of revised manuscript and Page S59, S70 of Supporting Information.

Comment 16: How to adhere the MXene nanolayer onto the glass shown in Fig. 1a, and how to separate the MXene nanolayer from the glass after the fabrication?

Response 16: We would like to clarify that the ps-MXene nanolayers were attached to polystyrene (PS) shrink films, and the glass slides adhered at the backside of shrink film. In other words, the textured ps-MXene nanolayer and the glass slides were attached to the two sides of a shrink film (see **Fig. S10**). After the ps-MXene-coated PS samples were immersed in a dichloromethane (DCM) bath, the middle PS shrink film was dissolved, and the M_n nanolayers were detached and became freestanding.

We have added more description in the revised manuscript to make this part clearer.

Fig. S10 Fabrication of freestanding M_n nanolayer.

Revision made

Page 6, 19 of revised manuscript Page S14 of Supporting Information.

Comment 17: *In movie S4, the avatar animations are sometimes ahead of the body motion, sometimes lag behind. Why? Avatar movement ahead of time is impossible and unreasonable.*

Response 17: We appreciate Reviewer #3's careful review, and we are also surprised to see that the avatar animation did move ahead of the full-body motion (specifically the squatting motions). Therefore, we have re-examined all the M_n sensor data carefully and provided our observation/explanation in the revised manuscript.

As shown in **Fig. S36** and **Movie S4**, the avatar's squatting movement (at the 22.6th second) was ahead of the video-recorded squatting motion (at the 23.0th second). The ahead motion determination was due to the early signals from the M_p sensor on the back waist (at the 22.4th second). In **Fig. S36**, before the squatting motion, the M_p sensor on the back waist (at the P8 joint) was able to sense the preparatory muscular stretching (at the 22.4th second), and the M_p sensor data were recognized as the early signals for a squatting motion. The early M_p sensor signals were observed every time during the repeated squatting motions, and the M_p sensor reached the peaks before the squatting motions were finished. Therefore, the CNN model would determine the avatar's squatting motions ahead of time (ca. 0.4 second). The ahead avatar animation was a clear evidence that the M_n sensors with high sensitivities and customized working windows are suitable for detecting delicate muscle movements.

Fig. S36 Signal timelines of a M_p sensor on the back waist and the location trajectories of P8 joint that were extracted from the recorded video or determined from the CNN model.

Revision made

Page 15 of revised manuscript Page S40, S69 of Supporting Information.

REVIEWERS' COMMENTS

Reviewer #1 (Remarks to the Author):

The authors have addressed all issues. This version is fine to be published.

Reviewer #3 (Remarks to the Author):

The reviewer satisfies the revision and supports the publication of the revised manuscript.

Responses to the Reviewers' Comments

Reviewer #1 (Remarks to the Author):

Comment: The authors has addressed all issues. This version is fine to be published.

Response: Thank Reviewer #1 so much for his/her great efforts in reviewing our revised manuscript.

Reviewer #3 (Remarks to the Author):

Comment: The reviewer satisfies the revision and supports the publication of the revised manuscript.

Response: Thank Reviewer #3 so much for his/her great efforts in reviewing our revised manuscript.